# Astrocytes detect and upregulate transmission at inhibitory synapses of somatostatin interneurons onto pyramidal cells

Marco Matos [1,2], Anthony Bosson[1,2], Ilse Riebe[1,2], Clare Reynell[1,2], Joanne Vallée[1,2], Isabel Laplante[1,2], Aude Panatier[3,4], Richard Robitaille[1,2] & Jean-Claude Lacaille [1,2]

Astrocytes are important regulators of excitatory synaptic networks. However, astrocytes regulation of inhibitory synaptic systems remains ill defined. This is particularly relevant since GABAergic interneurons regulate the activity of excitatory cells and shape network function. To address this issue, we combined optogenetics and pharmacological approaches, two-photon confocal imaging and whole-cell recordings to specifically activate hippocampal somatostatin or paravalbumin-expressing interneurons (SOM-INs or PV-INs), while monitoring inhibitory synaptic currents in pyramidal cells and $Ca^{2+}$ responses in astrocytes. We found that astrocytes detect SOM-IN synaptic activity via $GABA_BR$ and GAT-3-dependent $Ca^{2+}$ signaling mechanisms, the latter triggering the release of ATP. In turn, ATP is converted into adenosine, activating $A_1Rs$ and upregulating SOM-IN synaptic inhibition of pyramidal cells, but not PV-IN inhibition. Our findings uncover functional interactions between a specific subpopulation of interneurons, astrocytes and pyramidal cells, involved in positive feedback autoregulation of dendritic inhibition of pyramidal cells.

---

[1] Département de Neurosciences, Faculté de Médecine, Université de Montréal, PO Box 6128, Station Centre-Ville, Montreal, QC H3C 3J7, Canada. [2] Groupe de Recherche sur le Système Nerveux Central, Université de Montréal, PO Box 6128, Station Centre-Ville, Montreal, QC H3C 3J7, Canada. [3] Neurocentre Magendie, Inserm U1215, 33077 Bordeaux, France. [4] Université de Bordeaux, 33077 Bordeaux, France. These authors contributed equally: Marco Matos, Anthony Bosson. These authors jointly supervised this work: Richard Robitaille, Jean-Claude Lacaille. Correspondence and requests for materials should be addressed to R.R. (email: richard.robitaille@umontreal.ca) or to J.-C.L. (email: jean-claude.lacaille@umontreal.ca)

nformation processing in the hippocampus relies on an intricate circuit of excitatory projection cells and local inhibitory inter-neurons, where interneurons orchestrate the pattern of excitation and synchronization of the neuronal network[1]. Additionally, astrocytes regulate transmission in hippocampal circuits through bidirectional communication with neurons. This intimate structural and functional interaction between astrocyte, pre-synaptic terminal and postsynaptic cell, termed "tripartite synapse", proposes that astrocytes sense synaptic activity through membrane receptors, which leads to increased intracellular $Ca^{2+}$ levels, triggering glio-transmitter release[2]. Gliotransmitters, in turn, act on neurons regulating their synaptic and extrasynaptic activity, enabling temporal and spatial integration of information[2]. Mounting evidence demonstrated that astrocyte-derived purines adjust synaptic efficacy to the needs of the particular network. For example, ATP released by hippocampal astrocytes, is converted extracellularly into adenosine, which acts on presynaptic adenosine $A_1$ receptors ($A_1R$), established inhibitors of excitatory transmission[3–8] and involved in heterosynaptic depression[3,6]. This important mechanism participates in sleep regulation[9] and hippocampus-related cognition[10]. Conversely, purinergic signaling in astrocytes increases basal excitatory transmission through activation of facilitatory $A_{2A}$ receptors ($A_{2A}R$)[11]. Thus, hippocampal astrocytes use a balance of $A_1R$–$A_{2A}R$ activation to bidirectionally modulate synaptic plasticity and influence cognitive processes.

While many studies investigated astrocyte modulation of excitatory components of synaptic networks, the involvement of astrocytes at inhibitory synapses is still largely undefined[12]. Astrocytes respond to exogenous GABA application[12] but also to endogenous GABAergic activity with $Ca^{2+}$ oscillations via several mechanisms, including $GABA_A$ receptors ($GABA_A$Rs)[13], $GABA_B$ receptors ($GABA_B$Rs)[3,14–16], and GABA transporters (GATs)[13,17,18]. Such endogenous activation of GABA receptors and transporters in astrocytes evokes astrocytic release of glutamate[14,19,20] or ATP[3], efflux of chloride[13] and alterations in GATs activity[21–23], processes that can modulate neuronal activity. Interestingly, sustained depolarization of astrocytes producing intracellular $Ca^{2+}$ increases potentiates miniature inhibitory postsynaptic currents (mIPSCs) in hippocampal pyramidal cells[14]. Also, reduction of astrocyte resting $Ca^{2+}$ levels mediated by TRPA1 cation channels decreases inhibitory synaptic responses in interneurons by reducing GAT-3-mediated GABA transport[24]. However, it lacked effect at pyramidal cell inhibitory synapses, suggesting modulatory mechanisms specific to some inhibitory synapses in hippocampal networks. Indeed, highly compartmentalized inhibitory synapses onto hippocampal pyramidal cells originate from heterogeneous interneuron subtypes[1,25] and it remains to be determined how astrocytes influence interneuron-specific inhibitory synapses.

In the hippocampus, pyramidal cell dendritic regions are densely populated by astrocytes with fine astrocytic processes surrounding dendrites and contacting a large proportion of synapses[26,27]. We demonstrated that astrocytic-driven hetero-synaptic depression occurred at excitatory synapses on pyramidal cell apical dendrites[3]. However, pyramidal cells also receive a significant part of their inhibitory synapses in these dendritic regions[28]. Somatostatin-expressing interneurons (SOM-INs) are a major group of interneurons targeting pyramidal cell dendrites[28,29]. SOM-INs regulate synaptic integration, dendritic burst firing and synaptic plasticity of pyramidal cells, and play a crucial role in hippocampal-dependent contextual fear learning[30–33]. In contrast, another major type of interneurons, parvalbumin-expressing interneurons (PV-INs), target the peri-somatic domain of pyramidal cells[28]. PV-INs control spike timing of pyramidal cells and are essential for spatial working memory[31,34]. In addition, it has been recently demonstrated that

astrocytes in neocortex are differentially affected by optogenetic activation of interneurons. SOM-INs activation results in robust $GABA_B$ receptor-mediated $Ca^{2+}$ elevations in astrocytes whereas PV-INs activation induces weak $Ca^{2+}$ elevations[35]. Thus, SOM-IN and PV-IN synapses onto pyramidal cell are interesting potential targets for astrocyte regulation. To address this question, we used cell-specific expression of channelrhodopsin-2 in SOM-INs or PV-INs[36], whole-cell recordings from pyramidal cells, 2-photon $Ca^{2+}$ imaging in astrocytes, and pharmacological approaches to examine astrocyte interactions at SOM-IN and PV-IN inhibitory synapses onto pyramidal cells. We found an endogenous mechanism of astrocyte-mediated upregulation of SOM-IN, but not PV-IN, inhibitory synapses onto pyramidal cells as revealed by the blockade of GAT-3 activity, inhibition of $Ca^{2+}$ signaling in astrocytes and prevention of the extracellular conversion of ATP to adenosine or $A_1$Rs activation. Our findings suggest a cell-specific interaction between SOM-INs, astrocytes, and pyramidal cells responsible for positive feedback auto-regulation of dendritic inhibition of hippocampal pyramidal cells.

## Results

**$A_1R$ upregulates inhibition of pyramidal cells by SOM-INs.** We examined the implication of astrocytes at dendritic inhibitory synapses onto CA1 pyramidal cells using a cell-specific optoge-netic approach with Cre-dependent expression of channelrho-dopsin 2 (ChR2) in dendrite-projecting Cre-expressing somatostatin interneurons (SOM-INs). We used SOM-ChR2/EYFP transgenic mice, in combination with whole-cell recording of inhibitory postsynaptic currents (IPSCs) in CA1 pyramidal cells in acute hippocampal slices (Fig. 1a, b). Graded optogenetic stimulation of SOM-INs (light pulse duration 0.5–5 ms; 0.1 Hz) elicited gradually increasing depolarization and firing (1–2 action potentials) in current-clamp recordings from SOM-INs (Supplementary Figure 1a). The same optogenetic stimulation of SOM-INs evoked $GABA_A$R-mediated IPSCs (SOM-IPSCs) of increasing amplitude in pyramidal cells (Supplementary Figures 1c and 1e).

We first determined whether adenosine, resultant from catabolism of ATP released by astrocytes, and acting on $A_1$Rs could regulate synaptic inhibition by SOM-INs of pyramidal cells. We bath-applied the selective $A_1R$ antagonist DPCPX (100 nM) to block endogenous adenosine activation of $A_1$Rs. Optogenetic stimulation of SOM-INs evoked SOM-IPSCs in CA1 pyramidal cells that remained stable during vehicle application and after washout (Fig. 1c, h). In contrast, SOM-IPSC amplitude decreased during application of DPCPX (64.4 ± 7.4% of control Fig. 1h), which reversed upon washout (Fig.1d). These results suggest that SOM-IPSCs are upregulated by endogenous adenosine activating $A_1$Rs. This was not due to DPCPX effect on SOM-INs and decreasing their response to optogenetic stimulation since responses of SOM-INs to optogenetic stimulation (Fig.1g) were similar in control, DPCPX and after washout (action potential number: 1.75 ± 0.25; 1.67 ± 0.33, and 1.5 ± 0.30, respectively; $n = 4$ cells; $p > 0.05$). These data indicate that DPCPX directly modulates SOM-IPSCs, suggesting that endogenous adenosine acting on $A_1$Rs upregulates SOM-IN IPSCs in pyramidal cells.

We next tested whether the adenosine involved in $A_1R$ modulation of SOM-IPSCs is a product of breakdown of extracellular ATP, presumably released from astrocytes[3–10]. We prevented the extracellular catabolism of ATP into adenosine by using AMP-CP (200 μM), an inhibitor of CD73/ecto-5′-nucleoti-dase that converts 5′-AMP into adenosine. AMP-CP reversibly decreased SOM-IPSC amplitude (57.1 ± 10.6% of control Fig. 1e, h). Application of AMP-CP after a prior application of DPCPX failed to further decrease SOM-IPSCs (66.5 ± 5.9% of control Fig. 1f, i). These results suggest that the source of adenosine producing

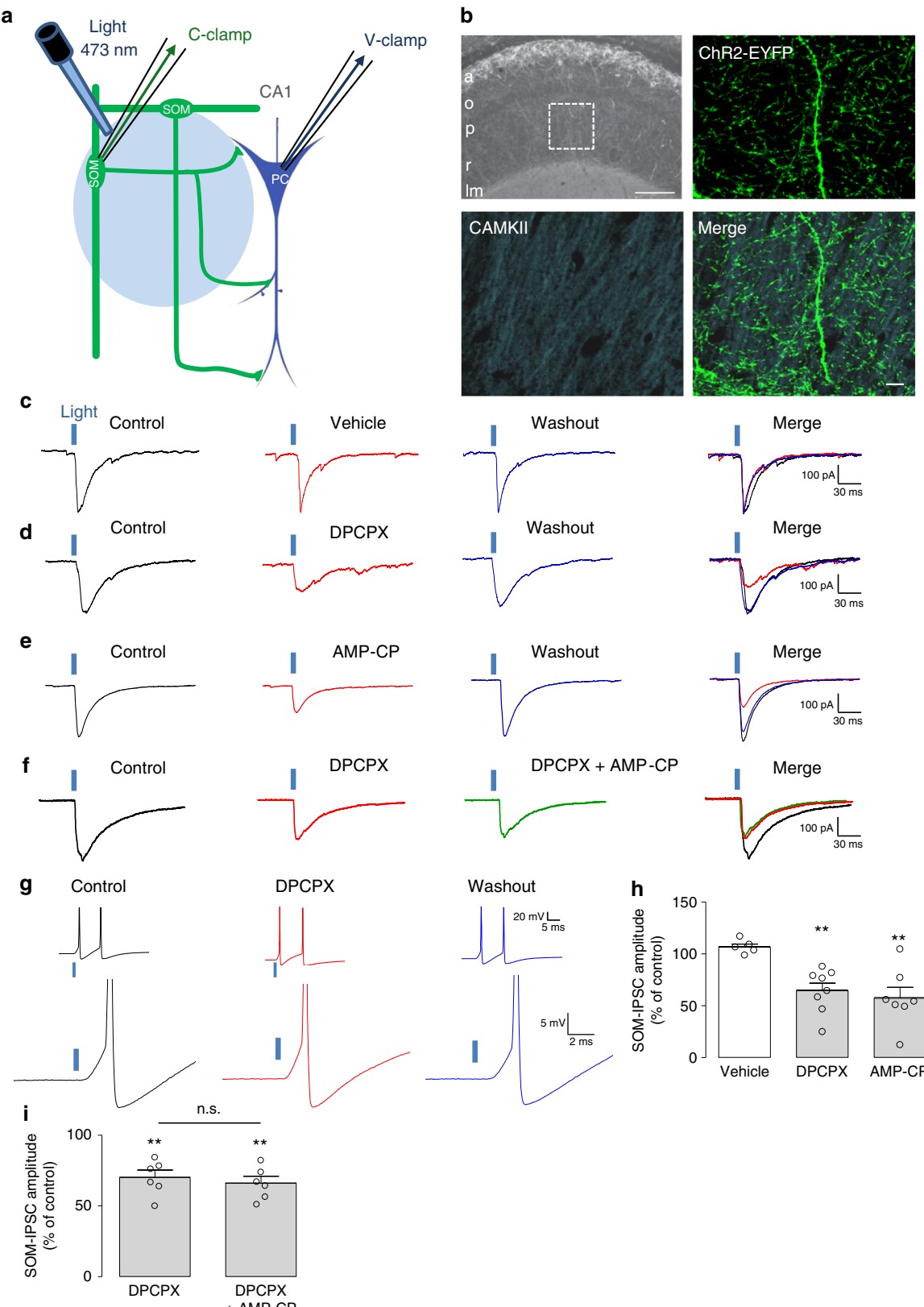

endogenous activation of $A_1Rs$ and upregulation of SOM-IPSCs originates from the breakdown of extracellular ATP.

**Astrocyte $Ca^{2+}$ signaling upregulates inhibition by SOM-INs.** Previous work suggested that astrocyte $Ca^{2+}$ signaling triggers ATP release leading to adenosine formation and activation of

$A_1Rs^{3-10}$. Therefore, we explored the role of astrocyte $Ca^{2+}$ signaling in the upregulation of SOM-IN inhibition. We used intracellular dialysis of the $Ca^{2+}$ chelator BAPTA to impair astrocyte $Ca^{2+}$ responses (as previously[3,11]) and examined the effect on inhibition of pyramidal cells by SOM-INs (Fig. 2a).

**Fig. 1** Endogenous activation of $A_1$Rs by adenosine upregulates inhibition evoked in pyramidal cells by SOM-IN optogenetic stimulation. **a** Diagram of experimental arrangement with optogenetic stimulation of SOM-INs (SOM) expressing ChR2-EYFP and whole-cell recordings of pyramidal cells (PC) and SOM-INs. **b** Top left: low-magnification fluorescence microscopy image of the hippocampal CA1 area from SOM-ChR2/EYFP mice, with SOM-INs (in white) somas in strata alveus (a) and oriens (o) and multiple axonal projections to the strata pyramidale (p), radiatum (r) (box) and lacunosum-moleculare (lm). Top right and bottom: representative immunofluorescence images of CaMK-II labeling of PCs (blue) and ChR2-EYFP labeling of SOM-INs (green) in *stratum radiatum* of SOM-ChR2/EYFP mice. Scale bars 100 μm (top-left), 20 μm (bottom-right). **c** Representative voltage-clamp traces showing unchanged SOM-IPSCs evoked in a PC by optogenetic stimulation before (control; left, black), 20 min after vehicle application (0.01% DMSO; middle, red) and 30 min after washout (right, blue). **d** Representative traces showing the reversible reduction of SOM-IPSC amplitude after 20 min application of DPCPX (100 nM; red). **e** Representative traces illustrating the reversible reduction of SOM-IPSC amplitude after 20 min application of AMP-CP (200 μM; red). **f** Representative traces of an occlusion experiment where reduction of SOM-IPSCs by DPCPX (red) prevents further reduction of SOM-IPSCs by additional application of AMP-CP (green). **g** Representative current-clamp traces from a ChR2/EYFP-expressing SOM-IN showing the unchanged depolarization and action potential firing evoked by optogenetic stimulation before (left, black), during DPCPX application (middle, red) and after washout. The traces above reveal that two action potentials are elicited per single stimulation, in all conditions. The panel below with clipped action potentials for each condition shows the unchanged depolarization level triggering the action potentials. **h** Summary bar graph depicting changes in amplitude of SOM-IPSCs in PCs. DPCPX ($n = 8$) and AMP-CP ($n = 7$) significantly decreased the amplitude of SOM-IPSCs, but vehicle ($n = 8$) did not. **i** Summary bar graph showing that, additional application of AMP-CP following application of DPCPX failed to further reduce SOM-IPSCs ($n = 6$). n.s. non-significant; **$p < 0.01$ (see Supplementary Table 1 for detailed statistical tests)

Astrocytes in *stratum radiatum* were identified by labeling with 0.25 μM sulforhodamine 101 red fluorescent dye (SR101, 0.25 μM), as described previously[37,38]. We verified that SR101 labels astrocytes (Supplementary Figure 2a) without affecting membrane properties and spontaneous IPSCs in pyramidal cells (Supplementary Figure 2c), as previously suggested for EPSCs[39] using a higher concentration.

During whole-cell recording from pyramidal cells, an astrocyte in *stratum radiatum* was contacted with a patch electrode containing low (0.1 mM) or high (20 mM) concentrations of BAPTA in cell-attached configuration. SOM-IPSCs were evoked by optogenetic stimulation of SOM-INs, keeping the astrocyte membrane intact to prevent BAPTA diffusion into the cell. Then, whole-cell configuration was established to enable BAPTA diffusion into the astrocyte. The patch pipette also contained Alexa 488 (100 μM) to visualize the astrocyte syncytium (~10–15 Alexa Fluor 488-labeled astrocytes covering an area ~100 μm in diameter) (Fig. 2b). After 20 min of astrocyte dialysis with 20 mM BAPTA, SOM-IPSCs were reduced ($57.2 \pm 4.4\%$ of control, Fig. 2d–e). Similar dialysis of astrocytes with low concentration (0.1 mM) of BAPTA did not alter SOM-IPSCs ($98.0 \pm 9.0\%$ of control, Fig. 2c, e). These results indicate that endogenous $Ca^{2+}$ activity in astrocytes upregulates SOM-IN inhibition of pyramidal cells.

**SOM-IN activation evokes $Ca^{2+}$ signals in astrocytes**. Since astrocyte $Ca^{2+}$-dependent processes upregulate SOM-IN inhibition of pyramidal cells, we next examined whether astrocytes respond to SOM-IN synaptic activity with $Ca^{2+}$ changes by analyzing $Ca^{2+}$ responses elicited in astrocytes by optogenetic stimulation of SOM-INs (Fig. 3a).

SR101-positive astrocytes in *stratum radiatum* were recorded in whole-cell current-clamp with pipettes containing CaSiR-1 (100 μM), a near-infrared $Ca^{2+}$ indicator with a maximum light-absorption spectrum (~650 nm) distinct from ChR2 (~473 nm)[40,41] (Fig. 3b). $Ca^{2+}$ transients were elicited in astrocyte processes by optogenetic stimulation of SOM-INs (trains of 5 ms pulses at 1 Hz for 5 s, optimal stimulation described in Supplementary Figure 3a-b) (Fig. 3c). SOM-INs optogenetic stimulation induced $Ca^{2+}$ transients in all astrocytic processes analyzed ($n = 26$; amplitude $27.2 \pm 2.6\%$ $\Delta F/F$; Fig. 3c, e). Optogenetic stimulation in slices from non-ChR2 expressing mice (SOM-Cre mice) did not elicit astrocyte $Ca^{2+}$ transients (Fig. 3d, $n = 4$). Whole-cell current-clamp recordings from YFP-expressing SOM-INs revealed that optogenetic stimulation

evoked on average 2 APs per pulse (Supplementary Figure 3 c, $n = 4$).

We tested the potential contribution of $GABA_B$Rs [3,14–16] and GAT-3[13,17,18] to SOM-IN-evoked $Ca^{2+}$ transients in astrocytes. First, we examined the distribution of GAT-3 and $GABA_B$Rs in relation to CA1 astrocytes using immunohistochemistry. As previously[24,42–44], GAT-3 immunoreactivity was localized in GFAP/S100β-positive astrocytic processes (Fig. 3f, g), with high levels in strata *pyramidale*, *radiatum*, and *lacunosum-moleculare* (Supplementary Figure 4). Similarly, $GABA_B$Rs were ubiquitous throughout CA1 region[45] (Supplementary Figure 4) and co-localized with GAT-3 on GFAP/S100β-positive astrocytic processes (Fig. 3f–g and Supplementary Figure 4). These results suggest that GAT-3 and $GABA_B$Rs co-localize in astrocyte processes in pyramidal cell dendritic area.

Next we tested the involvement of GAT-3 and $GABA_B$R in the astrocyte $Ca^{2+}$ transients evoked by optogenetic stimulation of SOM-INs with bath-application of the GAT-3-specific inhibitor (S)-SNAP-5114 (100 μM) and the selective $GABA_B$R antagonist CGP55845 (2 μM). Application of vehicle did not affect astrocyte $Ca^{2+}$ transient amplitude ($103.4 \pm 9.0\%$ of control, Fig. 3e, k) but (S)-SNAP-5114 decreased the amplitude of $Ca^{2+}$ transients ($47.7 \pm 4.0\%$ of control Fig. 3h, k). Application of CGP55845 also decreased the amplitude of astrocytic $Ca^{2+}$ responses ($71.0 \pm 1.2\%$ of control, Fig. 3i, k), but the reduction was smaller than the one induced by (S)-SNAP-5114 (Fig. 3k). Combined treatment with (S)-SNAP-5114 and CGP55845, to inhibit both GAT-3 and $GABA_B$R, had cumulative effects ($20.1 \pm 2.0\%$ of control Fig. 3j–k). Hence, optogenetic stimulation of SOM-INs induces $Ca^{2+}$ transients in astrocytes that are mediated predominantly via GAT-3 but also partially by $GABA_B$Rs.

**Astrocyte GAT-3 upregulates inhibition by SOM-INs**. Since both GAT-3 and $GABA_B$R were involved in astrocytic $Ca^{2+}$ responses evoked by SOM-INs stimulation, we evaluated whether GAT-3 and $GABA_B$R actions in astrocytes regulate inhibition of pyramidal cells by SOM-INs. While vehicle treatment (Fig. 4a) had no effect, application of (S)-SNAP-5114 reversibly decreased the amplitude of SOM-IPSCs evoked by optogenetic stimulation ($57.0 \pm 6.5\%$ of control Fig. 4b, d). In contrast, vehicle treatment (Fig. 4a) or CGP55845 application (Fig. 4c) did not affect SOM-IPSC amplitude ($107 \pm 5.0\%$ and $92.2 \pm 10.8\%$ of control, respectively, Fig. 4d). These results suggest that endogenous activation of GAT-3 (but not $GABA_B$R) upregulates inhibition by SOM-INs. Interestingly, GAT-3 and GFAP/S100β-positive

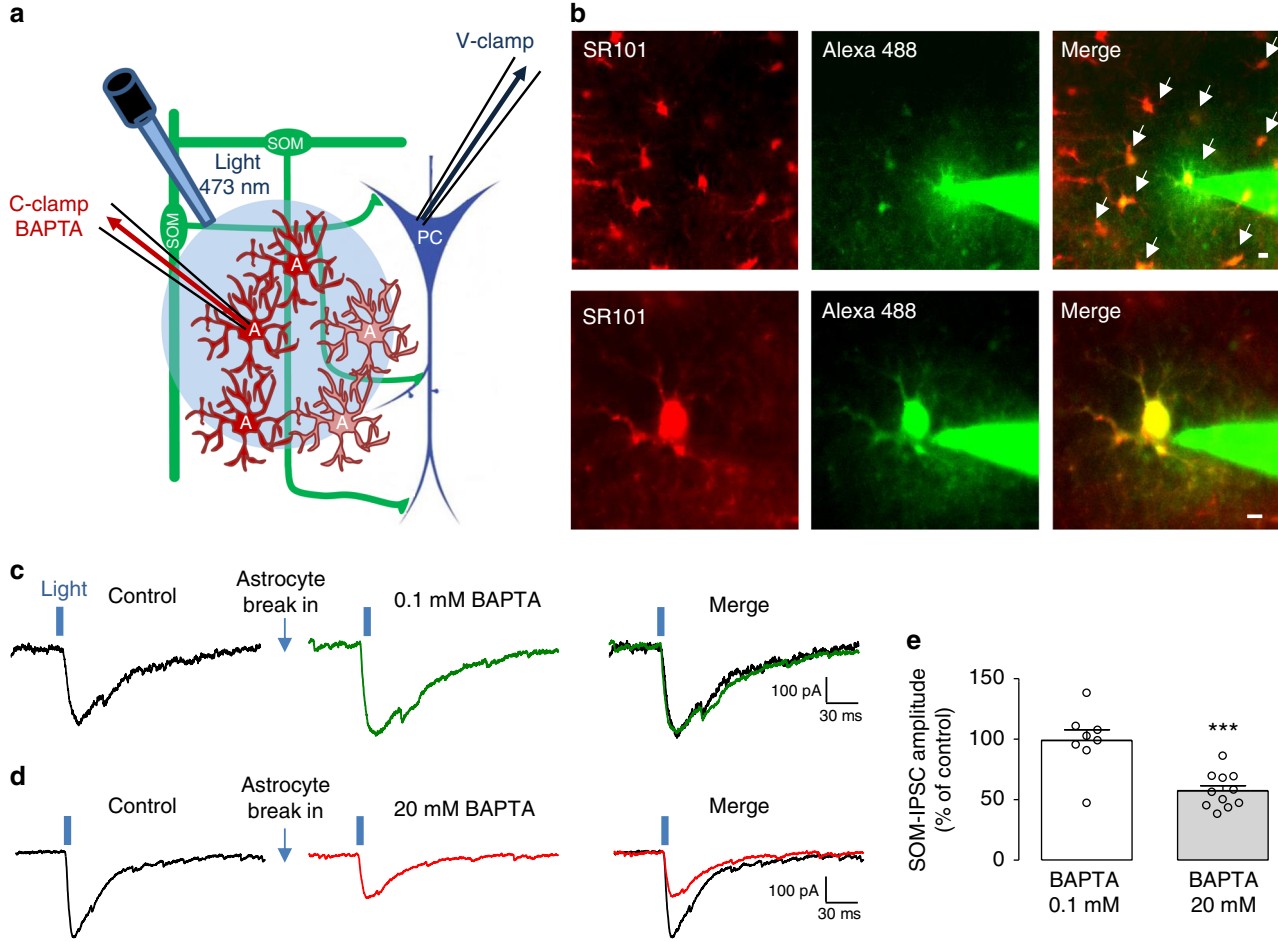

**Fig. 2** Blocking astrocyte $Ca^{2+}$ reduces pyramidal cell inhibition evoked by SOM-INs. **a** Diagram of experimental arrangement with optogenetic stimulation of SOM-INs, recording of SOM-IPSCs in pyramidal cells and dialysis of BAPTA in astrocytes via whole-cell patch pipette. **b** Confocal images (top, low-magnification; bottom, high-magnification) illustrating whole-cell recording from a SR101-labeled astrocyte (red) with Alexa 488 (100 μM) and BAPTA containing patch solution. Note the diffusion of Alexa 488 dye (green) to neighboring astrocytes (arrows in merged top image) following whole-cell recording from a single astrocyte. Scale bars 10 μm (top panel), 5 μm (bottom panel). **c** and **d** Representative traces showing SOM-IPSCs evoked in pyramidal cells by optogenetic stimulation. Responses were unchanged 20 min after astrocyte whole-cell break-in and dialysis of low concentration BAPTA (0.1 mM; green trace in **c**), and reduced after dialysis of higher concentration of BAPTA (20 mM; red trace in **d**). **e** Summary bar graph showing the effects on SOM-IPSCs of astrocyte dialysis with 0.1 mM ($n = 8$) or 20 mM ($n = 12$) BAPTA. A: astrocyte; PC: pyramidal cell; SOM: somatostatin interneuron. ***$p < 0.001$; (see Supplementary Table 1 for detailed statistical tests)

astrocytic processes in *stratum radiatum* were in close proximity to EYFP-labeled axonal projections of SOM-INs (Fig. 4e, f).

We next asked whether astrocyte $Ca^{2+}$ signaling was involved in the GAT-3 modulation of SOM-IPSCs using BAPTA dialysis in astrocytes prior to GAT-3 antagonist perfusion. As before, BAPTA dialysis decreased the amplitude of SOM-IPSCs (59.3 ± 4.0% of control Fig. 4h). However, (S)-SNAP- 5114 failed to further decrease SOM-IPSC amplitude after BAPTA dialysis (59.8 ± 3.7% of control Fig. 4g, h). These results indicate that BAPTA in astrocytes occluded the effect of GAT-3 blockade, suggesting that GAT-3 and $Ca^{2+}$ activity in astrocytes upregulated SOM-IN inhibition of pyramidal cells via a common mechanism.

**GAT-3 blockade occludes $A_1R$ modulation of SOM-IN inhibition.** Our results suggest that astrocyte GAT-3 activation leads to ATP release, activation of $A_1Rs$ and upregulation of SOM-IN inhibition of pyramidal cells. Next, we performed occlusion experiments to test this premise. We first applied GAT-3 inhibitor (S)-SNAP-5114 and observed a reduction in SOM-IPSC

amplitude (57.0 ± 4.0% of control Fig. 5a, c). Interestingly, application of the $A_1R$ antagonist DPCPX, failed to produce a further decrease in SOM-IPSCs (58.2 ± 3.3% of control Fig. 5a, c). These results suggest that prior blockade of GAT-3 prevents $A_1R$ modulation of SOM-IPSCs.

To further confirm this mechanism we tested if application of $A_1R$ agonist $N^6$-cyclopentyladenosine ($N^6$-CPA, 1 μM) during the blockade of GAT-3 could up-regulate inhibition by SOM-INs. In the presence of the GAT-3 inhibitor (S)-SNAP-5114 that decreased SOM-IPSC amplitude (59.3 ± 5.9% of control), application of the $A_1R$ agonist $N^6$-CPA increased SOM-IPSC amplitude (72.8 ± 5.9% of control Fig. 5b, d). This effect of $N^6$-CPA was blocked by application of the $A_1R$ antagonist DPCPX (Fig. 5b, d). However under basal conditions (in absence of inhibitors), application of $N^6$-CPA did not affect SOM-IPSC amplitude (99.0 ± 10.1% of control, Fig. 5f). Overall, these results are consistent with a GAT-3 activation of astrocytes leading to ATP release, activation of $A_1Rs$ and upregulation of SOM-IN inhibition (Fig. 5g).

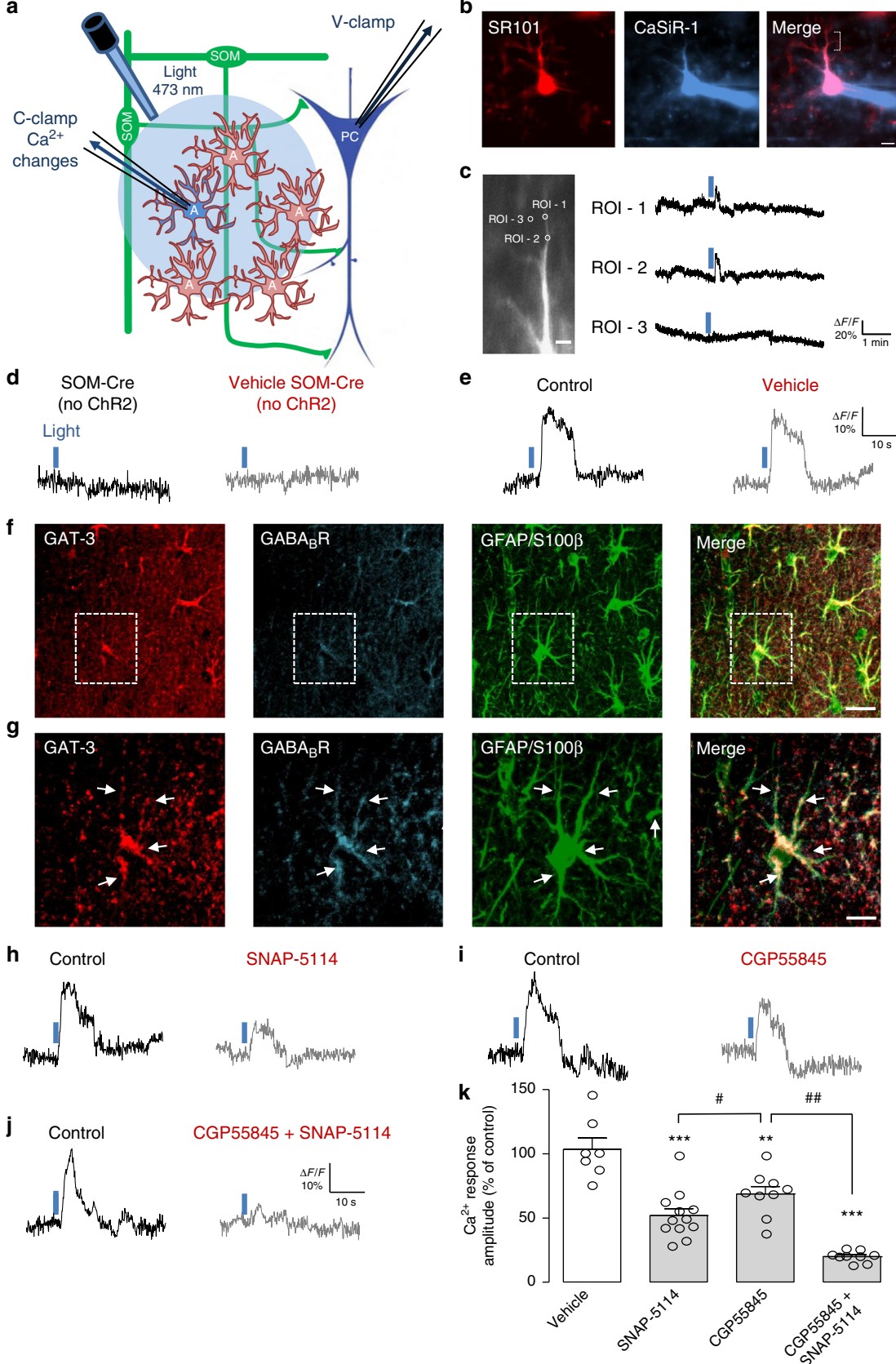

**Fig. 3** $Ca^{2+}$ responses in astrocytes evoked by SOM-INs are mediated by GAT-3 and $GABA_BRs$. **a** Experimental diagram with optogenetic stimulation of SOM-INs and $Ca^{2+}$ imaging in astrocytes during whole-cell recordings with the $Ca^{2+}$ indicator CaSiR-1 (100 μM) in CA1 hippocampal area of SOM-ChR2/EYFP mice. **b** Representative confocal images of SR101-positive astrocytes (red, left) filled with CaSiR-1 (blue, middle). The indicated region in the merged image (right) corresponds to the image in **c**. Scale bar 5 μm. **c** Left, fluorescence image of astrocyte processes with defined regions of interest (ROI) where changes in $Ca^{2+}$ levels were measured. ROI-1 and 2 are located along the astrocyte process, whereas ROI-3 is extracellularly for background measurement. Right, traces showing $Ca^{2+}$ changes (as percent changes in fluorescence relative to baseline fluorescence; %Δ*F/F*) in corresponding ROI evoked by optogenetic stimulation of SOM-INs. Scale bar 2 μm. **d** Representative traces from negative control experiments from SOM-Cre mice (without ChR2 expression) showing absence of $Ca^{2+}$ responses in astrocyte processes evoked by optogenetic stimulation. **e** Representative $Ca^{2+}$ signals in processes of astrocytes from SOM-ChR2/EYFP mice evoked by optogenetic stimulation of SOM-INs during control (black) and 20 min after application of vehicle (0.01% DMSO) (gray). **f** Representative z-stack of confocal immunofluorescence images of GFAP/S100β-positive astrocytes (green) with co-staining for $GABA_BR$ (blue) and GAT-3 (red) in the *stratum radiatum*. Scale bar 20 μm. **g** Higher magnification of region indicated in **f**. Scale bar 5 μm. **h**–**j** Representative $Ca^{2+}$ signals in processes of astrocytes from SOM-ChR2/EYFP mice evoked by optogenetic stimulation of SOM-INs during control (black traces) and 20 min after application of **h** (S)-SNAP-5114 (100 μM) (gray), **i** CGP55845A (2 μM) (gray), or **j** both CGP55845 and (S)-SNAP-5114 (gray). **k** Summary bar graph showing reduction in the amplitude of astrocyte $Ca^{2+}$ signals evoked by optogenetic stimulation by (S)-SNAP-5114 and CGP55845A. Experiments were conducted in the presence of AP-5 (20 μM), NBQX (10 μM), MPEP (25 μM), and Gabazine (5 μM). A: astrocyte, PC: pyramidal cell, and SOM: somatostatin interneuron. **$p < 0.01$, *** $p < 0.001$; #$p < 0.05$, ##$p < 0.01$ (see Supplementary Table 1 for detailed statistical tests)

**$A_1R$ and astrocyte GAT-3 do not regulate PV-IN inhibition**. We next examined if $A_1R$-mediated and GAT-3-mediated astrocytic modulation of synaptic inhibition of pyramidal cells also regulates inhibition by other interneuron types. We targeted ChR2 expression to PV-INs and recorded IPSCs evoked in CA1 pyramidal cells of PV-ChR2/EYFP transgenic mice by optogenetic stimulation (Fig. 6a, b). Graded optogenetic stimulation of PV-INs (light pulse duration 0.4–1 ms; 0.1 Hz) evoked $GABA_AR$-mediated IPSCs (PV-IPSCs) of increasing amplitude in pyramidal cells (Supplementary Figure 1d and 1f).

Next we used the same pharmacological approach to determine if endogenous activation of $A_1Rs$ regulates inhibition by PV-INs. Application of DPCPX (100 nM) failed to affect PV-IPSC amplitude (99.9 ± 6.0% of control, Fig. 6d, f), indicating that PV-IN inhibition of pyramidal cells is not subject to endogenous regulation by $A_1Rs$. Subsequently, we assessed if astrocytic GAT-3 activation regulates PV-IN inhibition using (S)-SNAP-5114. Application of (S)-SNAP-5114 (10 μM) did not change PV-IPSC amplitude (95.3 ± 9.1% of control Fig. 6e, f), showing that PV-IN inhibition of pyramidal cells is unaffected by the blockade of GAT-3. Thus, $A_1R$-mediated and GAT-3-mediated astrocytic regulation of synaptic inhibition of pyramidal cells may be specific to inhibition by SOM-INs.

**$A_1Rs$, GAT-3, and astrocyte $Ca^{2+}$ depress spontaneous IPSCs**. Synaptic inhibition of CA1 pyramidal cells originates from diverse interneurons[1,25,28]. Unlike IPSCs evoked by optogenetic stimulation of SOM-INs, spontaneous inhibitory postsynaptic currents (sIPSCs) in pyramidal cells reflect activation of inhibitory synapses originating from other types of interneurons[46,47]. Therefore, we examined whether $GABA_A$-mediated sIPSCs (Supplementary Figure 1g, 2b) were similarly regulated. Application of the $A_1R$ antagonist DPCPX (100 nM) led to a reversible increase in sIPSC amplitude (128.10 ± 6.0% of control Fig. 7a–c) and no change in frequency (97.14 ± 2.0% of control). This effect is the opposite of DPCPX actions on SOM-IPSCs (Fig. 1 d, h) but consistent with adenosine-mediated presynaptic depression at inhibitory synapses[48–51]. This suggests pathway-specific $A_1R$-mediated mechanisms differentially regulating inhibitory synapses from somatostatin and other interneurons.

We next tested the importance of extracellular ATP hydrolysis using application of the CD73/ecto 5' nucleotidase inhibitor (AMP-CP, 200 μM). It had no effect on sIPSC amplitude (99.00 ± 7.0% of control) or frequency (104.50 ± 5.0% of control) (Fig. 7d–f). This is in contrast to effects on SOM-IPSCs (Fig. 1e, h), implying that adenosine eliciting $A_1R$-mediated depression of

sIPSCs was not ATP-derived. This suggests that, unlike the modulation of SOM-INs inhibition, adenosine-mediated modulation of sIPSCs does not originate from ATP released from astrocytes.

Next we examined the actions of the GABA transporter GAT-3 on sIPSCs. In contrast to the inhibitory effects on SOM-IN inhibition (Fig. 4b, d), application of (S)-SNAP-5114 reversibly increased sIPSC amplitude (143.00 ± 13.0% of control) and frequency (120.10 ± 7.2% of control) (Fig. 7g, i). These facilitatory effects might arise from an increase in ambient levels of GABA due to blockade of the GABA transporter as previously observed[23,52], or to other GAT-3-dependent $Ca^{2+}$-mediated action of astrocytes on inhibitory synapses. To examine these possibilities, we blocked $Ca^{2+}$ signaling in astrocytes. BAPTA dialysis in astrocytes increased sIPSC amplitude (133.40 ± 3.2% of control) and frequency (117.00 ± 5.0% of control) (Fig. 7j, l). These facilitatory effects are opposite to the depressant effects of astrocyte BAPTA injections on SOM-IPSCs (Fig. 2d, e), suggesting pathway-specific astrocyte-mediated regulation of inhibitory synapses from somatostatin and other interneurons.

Finally, we tested if astrocyte $Ca^{2+}$ signaling was involved in the GAT-3 modulation of sIPSCs. Application of (S)-SNAP-5114 after BAPTA dialysis failed to further increase sIPSC amplitude (129.10 ± 3.0% of control) and frequency (119.54 ± 6.0% of control) (Fig. 7j, l). These results indicate that BAPTA in astrocytes occluded the effect of GAT-3 blockade, implying an endogenous suppression of spontaneous inhibitory synaptic activity by GAT-3-mediated $Ca^{2+}$ activity in astrocytes. These findings suggest differential actions of GAT-3-mediated $Ca^{2+}$ activity in astrocytes in the regulation of inhibitory synapses originating from somatostatin and other interneurons.

**Discussion**

Our findings reveal the existence of a dynamic endogenous mechanism by which astrocytes enhance SOM-IN inhibition of pyramidal cells, mediating a positive feedback autoregulation of dendritic inhibition of hippocampal pyramidal cells. We found that in situ CA1 hippocampal astrocytes sense endogenous GABA released by SOM-INs via GAT-3-mediated $Ca^{2+}$ signaling. Our data suggest that SOM-IN synaptic activity activates GAT-3-mediated $Ca^{2+}$ signaling in astrocytes, leading to ATP release and ensuing extracellular conversion into adenosine, followed by activation of $A_1Rs$ and enhancement of synaptic inhibition of pyramidal cells by SOM-INs. This astrocytic regulation appears specific to SOM-INs since inhibition of pyramidal cells by PV-INs is unaffected by $A_1R$ and GAT-3 blockade. In addition, our

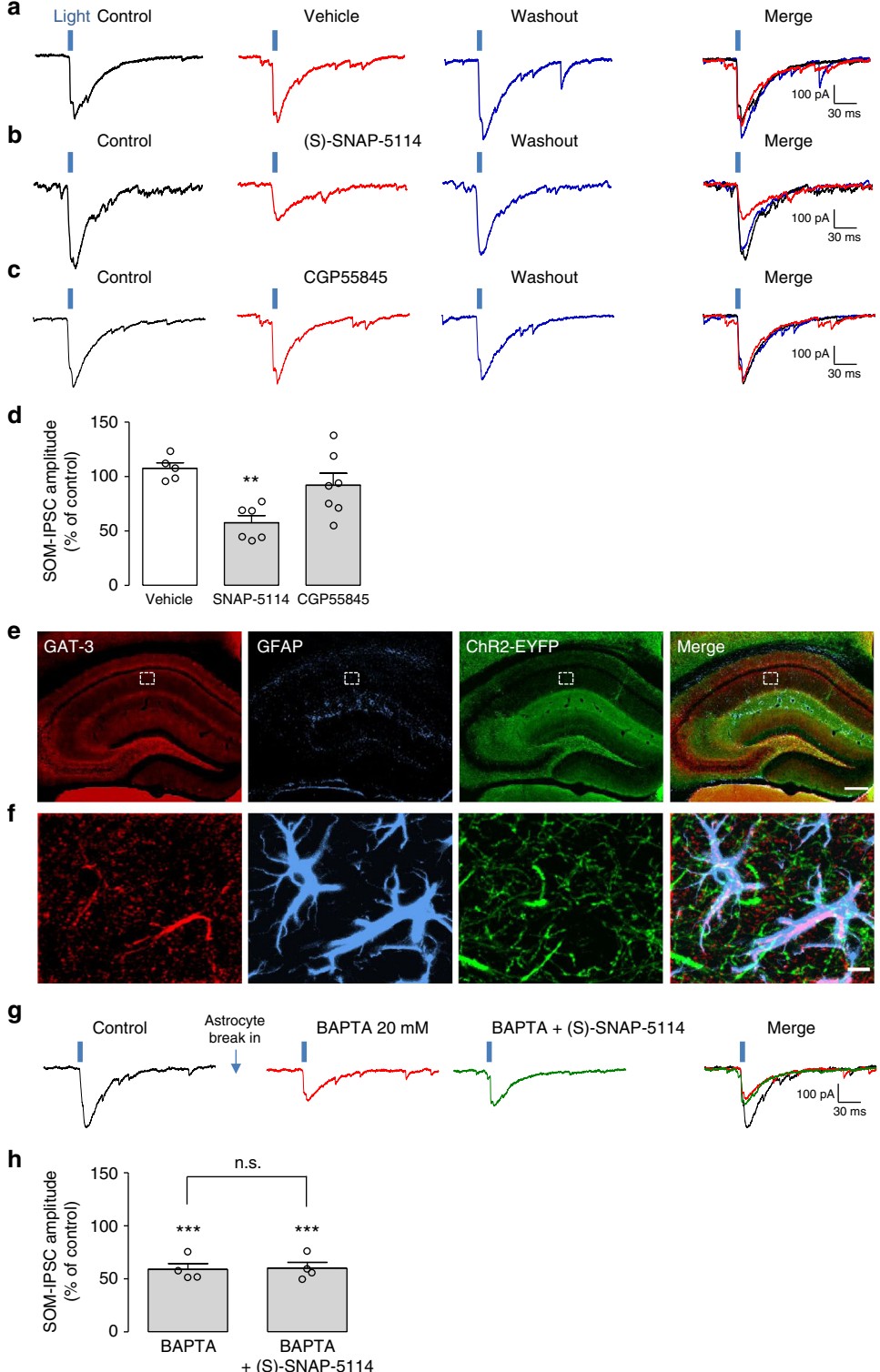

results show a different astrocyte-mediated modulation of spontaneous inhibitory responses in pyramidal cells, confirming a differential astrocytic regulation of inhibitory synapses made by SOM-INs and other types of interneurons on pyramidal cells. The endogenous astrocyte-mediated upregulation of SOM-IN inhibitory synapses on pyramidal cells provides evidence for a direct endogenous interaction between astrocytes, a specific subpopulation of inhibitory interneurons, and pyramidal cells which regulates hippocampal inhibitory synaptic transmission.

Previous work established that astrocytes, like neurons, are endowed with GABA$_A$ and GABA$_B$ receptors. Astrocyte GABA$_A$Rs contribute to morphological differentiation of astrocyte processes, whereas astrocyte GABA$_B$Rs participate in integration and modulation of neuronal activity[53]. Moreover, astrocytes express high-affinity GATs that remove the neurotransmitter from the synaptic cleft and limit spillover to neighboring synapses[54,55]. Of the four subtypes of GATs, GAT-3 is found exclusively in astrocytic processes in cortex and

**Fig. 4** Endogenous activation of GAT-3, but not GABA$_B$R, upregulates SOM-INs inhibition of pyramidal cells via astrocyte Ca$^{2+}$ signaling. **a–c** Representative traces of SOM-IPSCs from different pyramidal cells evoked by optogenetic stimulation in baseline condition (control—black traces), after 20 min in the presence of **a** vehicle (0.01% DMSO) (red trace), **b** GAT-3 blocker (S)-SNAP-5114 (100 μM, red trace), and **c** GABA$_B$R antagonist CGP55845 (2 μM, red trace), and after 30 min washout (blue traces), showing the reduction of SOM-IPSCs by (S)-SNAP-5114, but not vehicle or CGP55845. **d** Summary bar graph showing effects of (S)-SNAP-5114 application ($n = 6$), vehicle ($n = 7$) or CGP55845 ($n = 7$). **e** and **f** GAT-3 is present in astrocytes processes in close apposition to SOM-INs axonal projections. **e** Low-magnification immunofluorescence images of the hippocampus of SOM-ChR2/EYFP mice, depicting immunolabelling for GAT-3 (red), astrocyte-specific GFAP (blue), and ChR2-EYFP-positive SOM-INs labeling (green), with the merged images at right. The boxed region corresponds to the *stratum radiatum* area examined at higher magnification in **f**. Scale bar 100 μm. **f** Representative z-stack images, illustrating the close proximity between astrocyte-specific GAT-3 and GFAP labeling, and SOM-IN axonal projections in the *stratum radiatum*. Scale bar 5 μm. **g** Representative traces of an occlusion experiment illustrating that astrocyte dialysis with 20 mM BAPTA for 20 min reduces SOM-IPSCs (red trace) and prevents the reduction of SOM-IPSCs by 20 min application of (S)-SNAP-5114 (100 μM, green trace). **h** Summary bar graph illustrating that, following BAPTA dialysis in astrocytes, (S)-SNAP-5114 is unable to further reduce SOM-IPSCs ($n = 8$). **$p < 0.01$, ***$p < 0.001$, n.s. non-significant (see Supplementary Table 1 for detailed statistical tests)

hippocampus[42–44,54,55]. A role for GAT-3 in GABA uptake and regulation of GABA$_A$R-mediated inhibition has been suggested in studies with blockade of GAT-3 activity resulting in increases in phasic (IPSCs) and tonic inhibition[21–23]. However in hippocampus, such a role of GAT-3 in extracellular GABA regulation occurs only when GAT-1 function is prevented, or during excessive network activity and GABA release[23].

Astrocytic GAT-3[13,17,18] and GABA$_B$Rs[4,14–16] have also been implicated in Ca$^{2+}$ signaling in astrocytes. In particular, GABA-evoked Ca$^{2+}$ events in olfactory bulb astrocytes are fully prevented by GAT-3 blockers, only partially by GABA$_B$R antagonists and not affected by GABA$_A$R antagonists[17]. These observations are consistent with our findings that optogenetic stimulation of SOM-INs induced Ca$^{2+}$ transients in astrocytes via GAT-3 and GABA$_B$Rs (Fig. 3). Interestingly, our results highlight a key contribution of astrocytic GAT-3-mediated Ca$^{2+}$ signaling to upregulation of synaptic inhibition, as revealed by the blockade of GAT-3, but not GABA$_B$Rs, of SOM-IN evoked IPSCs in pyramidal cells (Fig. 4). As previously suggested[17,18], GAT-3-mediated Ca$^{2+}$ events could involve GAT-3 activation leading to inhibition of Na$^+$/Ca$^{2+}$ exchanger and subsequent Ca$^{2+}$-induced Ca$^{2+}$ release from internal stores. This may be an auto-regulated mechanism since astrocytic Ca$^{2+}$ signals can conversely modulate GAT-3 activity and protein levels[24]. The possibility of a coordinated transporter/receptor mechanism involving GAT-3 and GABA$_B$Rs in Ca$^{2+}$ transients in astrocytes is also possible, owing to their intimate co-localization in astrocytes (Fig. 3 and Supplementary Fig. 4) and a recent report showing that GABA$_B$Rs controls GAT-3 levels in astrocytes in vivo during synaptogenesis[56].

Previous work showed that activation of Schaffer collaterals evoked interneuron-mediated Ca$^{2+}$ signaling in astrocytes dependent on GABA$_B$R[3] and GAT-3[18] mechanisms, with subsequent ATP-derived adenosine formation, A$_1$R activation and heterosynaptic depression of excitatory transmission. In addition, astrocytes can increase Schaffer collateral excitatory transmission through the release of adenosine that activates facilitatory A$_{2A}$R receptors (A$_{2A}$R)[11], suggesting that hippocampal astrocytes use a balance of A$_1$R–A$_{2A}$R activation to bi-directionally modulate hippocampal excitatory synapses.

However, the direct contribution of astrocytes and adenosine signaling to GABAergic inhibitory activity remains under-explored[12]. An early observation by Nedergaard's group showed that large sustained depolarizations of astrocytes produced potentiation of miniature IPSCs in pyramidal cells, which was prevented by BAPTA dialysis in astrocytes[14]. A more recent publication showed that astrocyte Ca$^{2+}$ chelation did not affect mIPSCs in pyramidal cells but reduced mIPSCs in hippocampal *stratum radiatum* interneurons by interfering with GAT-3

function and increasing ambient GABA levels[23]. As a whole, these observations suggest that mIPSCs in pyramidal neurons are less susceptible to ambient GABA and that hippocampal astrocytes differentially regulate basal transmission at inhibitory synapses onto interneurons and pyramidal cells.

Conversely, whether astrocyte modulation of inhibitory synapses is specific to certain types of inhibitory interneurons synapses has not been fully established. The difficulty resides in part from the diverse nature of inhibitory transmission, with heterogeneous interneuron populations contacting pyramidal cells and acting on different sub-cellular compartments and time-windows[1,25,28]. Nevertheless, it has been recently shown that optogenetic activation of hippocampal astrocytes increases the firing frequency of cholecystokinin-expressing interneurons (CCK-INs), but not PV-INs via ATP release and decrease pyramidal cells excitability via adenosine[57]. Our data show that astrocytes differentially sense endogenous synaptic activity at PV-IN and SOM-IN synapses to, in turn, increase the efficacy at SOM-IN synapses on pyramidal cells. This is indicative of a specific communication between a particular subpopulation of interneurons and astrocytes involvement in the positive feedback autoregulation of dendritic inhibition of pyramidal cells.

We demonstrated that BAPTA dialysis into astrocytes differentially, and bi-directionally, modulates SOM-IN-evoked IPSCs (Fig. 2) and spontaneous IPSCs (Fig. 7), suggesting that endogenous astrocytic Ca$^{2+}$ signaling enhances inhibition of pyramidal cells by SOM-INs but reduces pyramidal cell inhibition by other interneuron populations. It is important to note that while somatic recordings of pyramidal cells can detect distant synaptic inhibitory currents along the complete somato-dendritic axis if evoked by stimulation (i.e. SOM-IPSCs), they can only detect spontaneous IPSCs generated at proximal somatic and dendritic synapses[46,47]. Because of this intrinsic technical limitation, the sIPSCs measured in our experiments most likely reflected activation of perisomatic synapses whereas the SOM-IPSCs mostly originated from dendritic synapses. This implies that the different effects of GAT-3-mediated Ca$^{2+}$ activity in astrocytes, and A$_1$R modulation, on SOM-IN-evoked IPSCs and sIPSCs are due to selective regulation of inhibitory synapses originating from different types of interneurons. Hence, our findings suggests the existence of pathway-specific functional interactions of astrocytes with different types of interneuron inhibitory synapses onto pyramidal cells, emphasizing the need to carefully distinguish between the different components of inhibitory circuits to identify astrocyte function at inhibitory synapses[1,25,28]. Moreover, our results suggest that by differently regulating diverse forms of inhibition, astrocytes may exert multiple functions in the regulation of synaptic integration along the somato-dendritic axis of pyramidal cells.

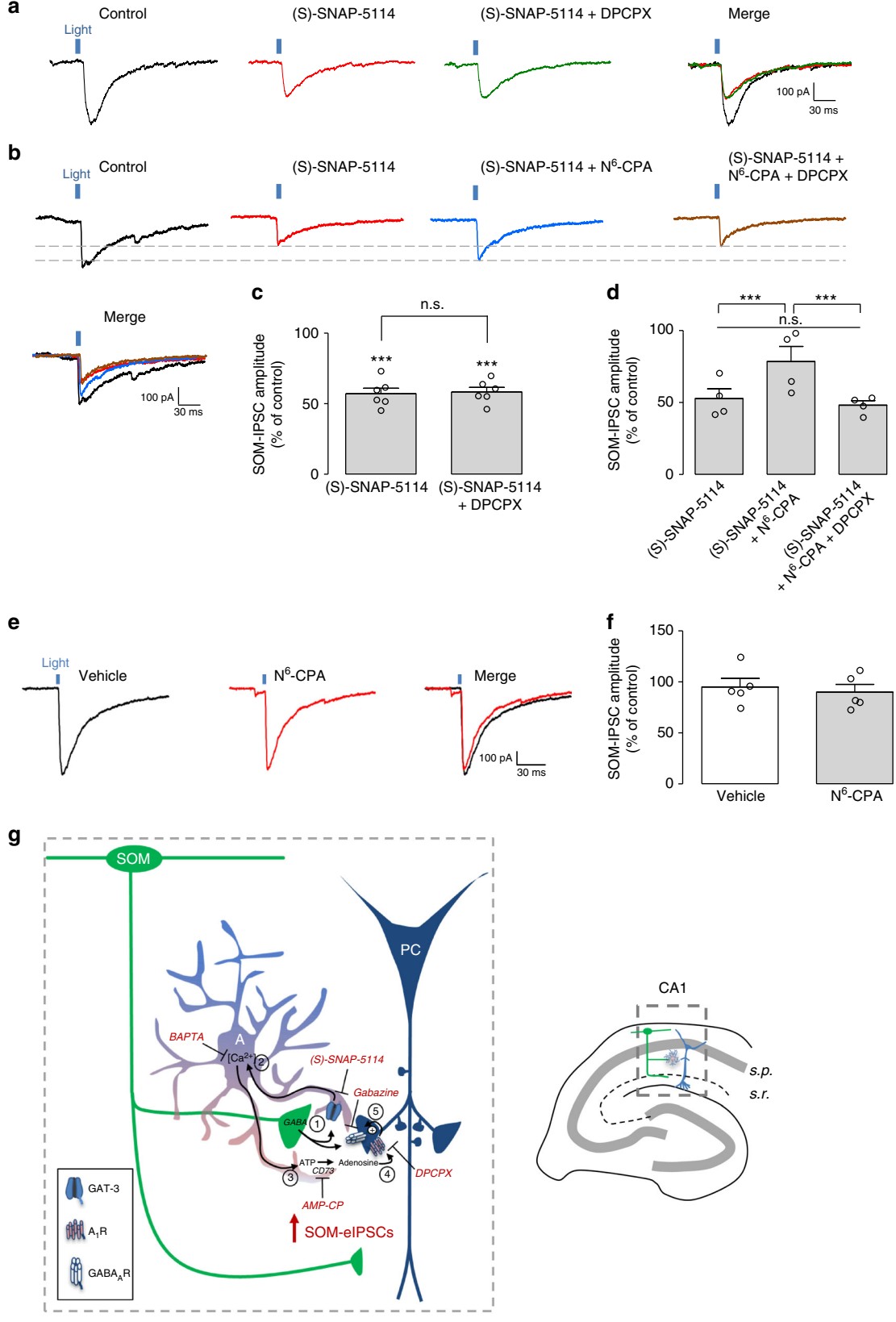

A potential problem with the BAPTA experiments is if BAPTA spread to gap junction-connected neighboring astrocytes it could as well leak to the extracellular space via hemichannels. Thus, in experiments with the high concentration of BAPTA (20 mM), leakage could impact on extracellular $Ca^{2+}$ and hence synaptic transmission. However, we have previously carried out experiments with a pipette containing BAPTA in the extracellular space to rule out potential effects of BAPTA leakage[3,11]. Moreover, the increased sIPSC amplitude in Fig. 7j–l argues against such an effect.

**Fig. 5** GAT-3 inhibition prevents $A_1R$ modulation of pyramidal cell inhibition by SOM-INs. **a** Representative traces of an occlusion experiment showing that reduction of SOM-IPSCs by (S)-SNAP-5114 (red trace) prevents further reduction of SOM-IPSCs by additional application of DPCPX (100 nM, green trace; see Fig. 1d for comparison). **b** Representative traces showing that the reduction of SOM-IPSCs following (S)-SNAP-5114 (red trace) application is significantly reversed by additional application of $A_1R$ agonist $N^6$-CPA (1 μM, blue trace). This reversal is blocked by application of DPCPX (brown trace). **c** Summary bar graph showing that additional application of the $A_1R$ antagonist DPCPX following application of the GAT-3 inhibitor (S)-SNAP-5114 failed to further reduce SOM-IPSCs ($n = 6$). **d** Summary bar graph showing reversion of SOM-IPSCs reduction by application of $N^6$-CPA following application of (S)-SNAP-5114 ($n = 4$). **e** Representative traces showing the absence of effect of $N^6$-CPA (1 μM red trace) alone on SOM-IPSCs. **f** Summary bar graph of $N^6$-CPA application ($n = 5$). **g** Summary diagram of the current model of astrocyte involvement in a positive feedback autoregulation of dendritic inhibition of CA1 pyramidal cells by SOM-INs. (1) SOM-INs release GABA at presynaptic axon terminals, which is taken up by GAT-3 transporter into astrocytes. (2) Increases in GAT-3 activity result in a consequent increase in astrocytic $Ca^{2+}$, (3) leading to astrocytic release of ATP and its extracellular catabolism by an ectonucleotidase cascade terminated by a final step of conversion into adenosine by ecto-5'-nucleotidase (CD73). (4) Adenosine activates postsynaptic $A_1Rs$ on pyramidal cell dendrites, which (5) enhance SOM-IN evoked-IPSCs by a mechanism leading to increased gain of function of postsynaptic $GABA_ARs$. Pharmacological inhibitors are indicated in red with their respective molecular target. n.s. non-significant, ***$p < 0.001$ (see Supplementary Table 1 for detailed statistical tests)

SOM-INs are a major interneuron subgroup[1,29] with their axons targeting dendrites of pyramidal cells[33], as well as other interneurons in pyramidal cell dendritic areas[32]. Pharmacological and optogenetic experiments showed that CA1 SOM-INs suppress pyramidal cell firing rate and burst spiking evoked by stimulation in vitro[30] and during spatial mapping in vivo[31]. Moreover, SOM-INs are critically involved in hippocampal-dependent learning since silencing SOM-INs during fear learning was shown to impair long-term contextual memory[33]. Our findings that astrocytes modulate SOM-IN inhibition of pyramidal cells suggest that such astrocyte-mediated positive feedback autoregulation of dendritic inhibition of hippocampal pyramidal cells could be important for hippocampal-dependent memory. At another level of regulation, astrocytes are able to influence rhythmic firing of neurons[37,57], therefore it would also be relevant to investigate the relationship between astrocyte and inhibitory synapses in the modulation of rhythmic brain activities that are important for hippocampal functions.

The neuromodulator adenosine is known to regulate GABAergic activity via $A_1R$ activation in the hippocampus. Indeed, in normal[48] and pathological conditions[49] $A_1R$ activation indirectly depressed polysynaptic inhibition in hippocampal pyramidal cells via a presynaptic inhibition of excitatory inputs onto inhibitory cells, but was unable to directly affect $GABA_AR$-mediated monosynaptic inhibition in pyramidal cells. In addition, $A_1R$ activation was also shown to suppress tonic $GABA_AR$-mediated inhibition in pyramidal cells and CB1-expressing inhibitory interneurons[58]. Our findings that endogenous $A_1R$ activation suppresses spontaneous inhibitory responses in pyramidal cells and up-regulates SOM-IN, but not PV-IN, mediated-inhibition of pyramidal cells suggest that different subcellular pools of $A_1Rs$ may be responsible for the selective regulation of different inhibitory synapses on pyramidal cells[59]. However, the mechanisms responsible for these differential $A_1R$ actions remain to be identified. In hippocampal pyramidal neurons, $A_1Rs$ are present both presynaptically, where they inhibit neurotransmitter release through G-protein-coupled inhibition of voltage-dependent $Ca^{2+}$ channels, and postsynaptically, where activation leads to G-protein-dependent activation of inwardly rectifying $K^+$ channels, inhibition of voltage-dependent $Ca^{2+}$ channels and decreased excitability[48,51,59]. Since spontaneous IPSCs were recorded during blockade of glutamate transmission, $A_1R$-mediated presynaptic inhibition of excitatory afferents to interneurons is unlikely. Therefore, the observed decreases in spontaneous IPSCs mediated by $A_1R$ activation could result from a decrease in presynaptic GABA release from other interneuron subpopulations, as previously suggested[60]. Conversely, enhancement of SOM-IN evoked IPSCs by $A_1R$ activation could be due to

postsynaptic inhibition of adenylate cyclase, reduced PKA activity and increased postsynaptic $GABA_AR$ function. Similar mechanisms were suggested for the enhancement of inhibition following ischemia[49].

Our results uncover an endogenous and selective interaction between SOM-INs, astrocytes, and pyramidal cells involved in a positive feedback autoregulation of dendritic inhibition of pyramidal cells. Since we found that similar regulation is not present at PV-IN inhibitory synapses, it will be important to determine whether all astrocytes are able to respond to GABAergic synaptic activity, or if different astrocyte subpopulations respond selectively to activity of distinct GABAergic interneurons. Furthermore, it will be interesting to identify the multiple cellular mechanisms involved in the interaction between $GABA_BR$, GAT-3, and ATP-derived adenosine, and whether these differ in astrocytic interactions with different population of interneurons. Finally, understanding the significance of interneuron/astrocytes/pyramidal cell communication in the modulation of hippocampal-dependent cognitive processes, or in pathological conditions such as epilepsy, should prove interesting for understanding hippocampal function, and potentially unveiling novel astrocytes-dependent pathological mechanisms.

## Methods

**Mice.** All experiments were approved by and performed in accordance with guidelines for maintenance and care of animals of the Canadian Council of Animal Care and Université de Montréal. To express the light-gated ion channel channelrhodopsin-2 in SOM-INs and PV-INs, heterozygous SOM or PV-IRES-Cre-ChR2(H134R)/EYFP mice (SOM or PV-ChR2/EYFP) were obtained by crossing SOM-IRES-Cre mice (kindly provided by Z.J. Huang—Cold Spring Harbor Laboratory, Cold Spring Harbor, NY; JAX no. 013044)[36] or Pvalb$^{tm1(cre)}$$^{Arbr}$ (PV-Cre; Jackson Labs; JAX no. 008069) with ChR2(H134R)/EYFP Ai32 mice (Jackson Labs; JAX no. 012569). Experiments were performed on 1–2 months old mice of either sex.

**Slice preparation.** Transverse hippocampal slices were obtained from 4 to 8-week-old SOM-ChR2/EYFP or PV-ChR2/EYFP mice[36]. Animals were anesthetized with isoflurane and the brain was rapidly excised and placed in ice-cold choline-based cutting solution saturated with 95% $O_2$ and 5% $CO_2$ containing the following (in mM): 120 choline chloride, 3 KCl, 1.25 $NaH_2PO_4$, 26 $NaHCO_3$, 8 $MgCl_2$, 20 glucose, pH 7.4 and 295 mOsmol. A block of brain tissue containing the hippocampus was prepared and transverse hippocampal slices (300 μm thick) were cut on a vibratome (Leica VT1000S, Nussloch, Germany). Slices were transferred to oxygenated artificial CSF (ACSF) at $33 \pm 0.5$ °C containing the following (in mM): 130 NaCl, 2.5 KCl, 1.25 $NaH_2PO_4$, 26 $NaHCO_3$, 10 glucose, 1.3 $MgCl_2$, 2 $CaCl_2$, pH 7.3–7.4, and 305–310 mOsmol, and allowed to recover for at least 1 h before being placed in oxygenated ACSF at room temperature (RT). For experiments, the slices were transferred to a recording chamber where they were perfused (2.5 ml/min) with ACSF at 32–34 °C for the course of the experiment. The NMDA receptor antagonist AP-5 (20 μM) and the AMPA and Kainate receptor antagonist NBQX (10 μM) were present in the superfusate of all experiments. Slices were used for a maximum of 6 h after cutting.

**Cell identification.** CA1 pyramidal cells, astrocytes and SOM-INs and PV-INs were identified using an infrared camera (70 series; Dage-MTI, Michigan City, IN)

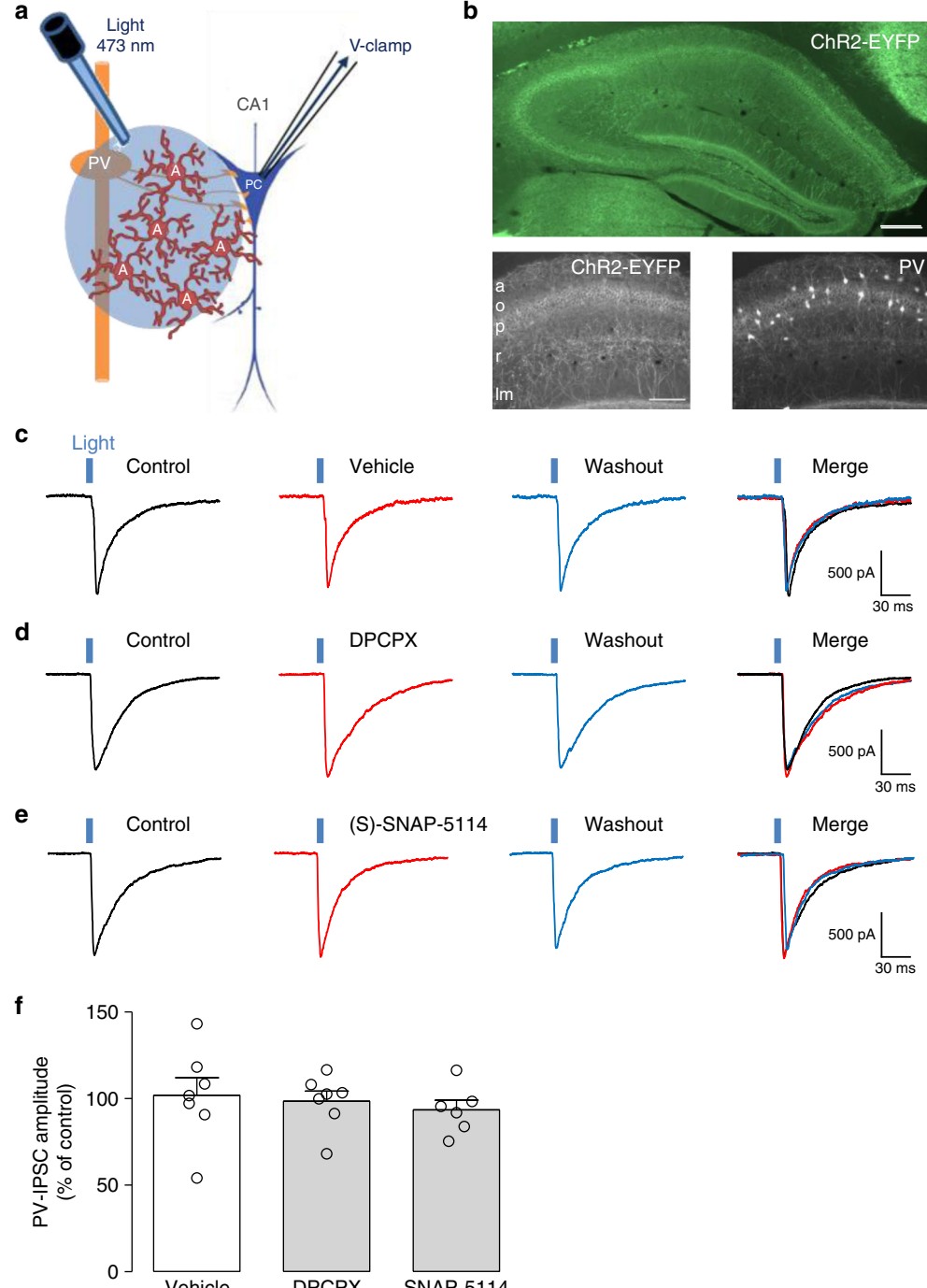

**Fig. 6** Endogenous activation of $A_1R$ and astrocytic GAT-3 do not regulate inhibition of pyramidal cells by PV-INs. **a** Diagram of experimental arrangement with selective optogenetic stimulation of PV-INs expressing ChR2-EYFP and whole-cell recordings of pyramidal cells (PC). **b** Top: low-magnification fluorescence microscopy image with green excitation filter of the hippocampus from PV-ChR2/EYFP mice. Scale bar 100 μm. Bottom left: higher magnification fluorescence image of ChR2-EYFP labeling of PV-INs in CA1 area. Bottom right: Paravalbumin immunostaining is strongest in and around *stratum pyramidale*. **c** Representative voltage-clamp traces showing unchanged PV-IPSCs evoked in pyramidal cells by optogenetic stimulation (blue vertical bar) before (control; left, black), 20 min after vehicle application (0.01% DMSO; middle, red) and 30 min after washout (right, blue). **d** Representative traces showing unchanged PV-IPSCs amplitude in pyramidal cells after 20 min application of the $A_1R$ antagonist DPCPX (100 nM, red). **e** Representative traces showing unchanged PV-IPSCs amplitude in pyramidal cells after 20 min application of the GAT-3 blocker (S)-SNAP-5114 (100 μM, red). **f** Summary bar graph depicting no significant change in the amplitude of PV-IPSCs in pyramidal cells. Vehicle ($n = 7$), DPCPX ($n = 7$) and (S)-SNAP-5114 ($n = 6$). PC pyramidal cell, PV paravalbumin interneuron, a *alveus*, o- *stratum oriens*, p *stratum pyramidale*, r *stratum radiatum*, lm *stratum lacunosum-moleculare* (see Supplementary Table 1 for detailed statistical tests)

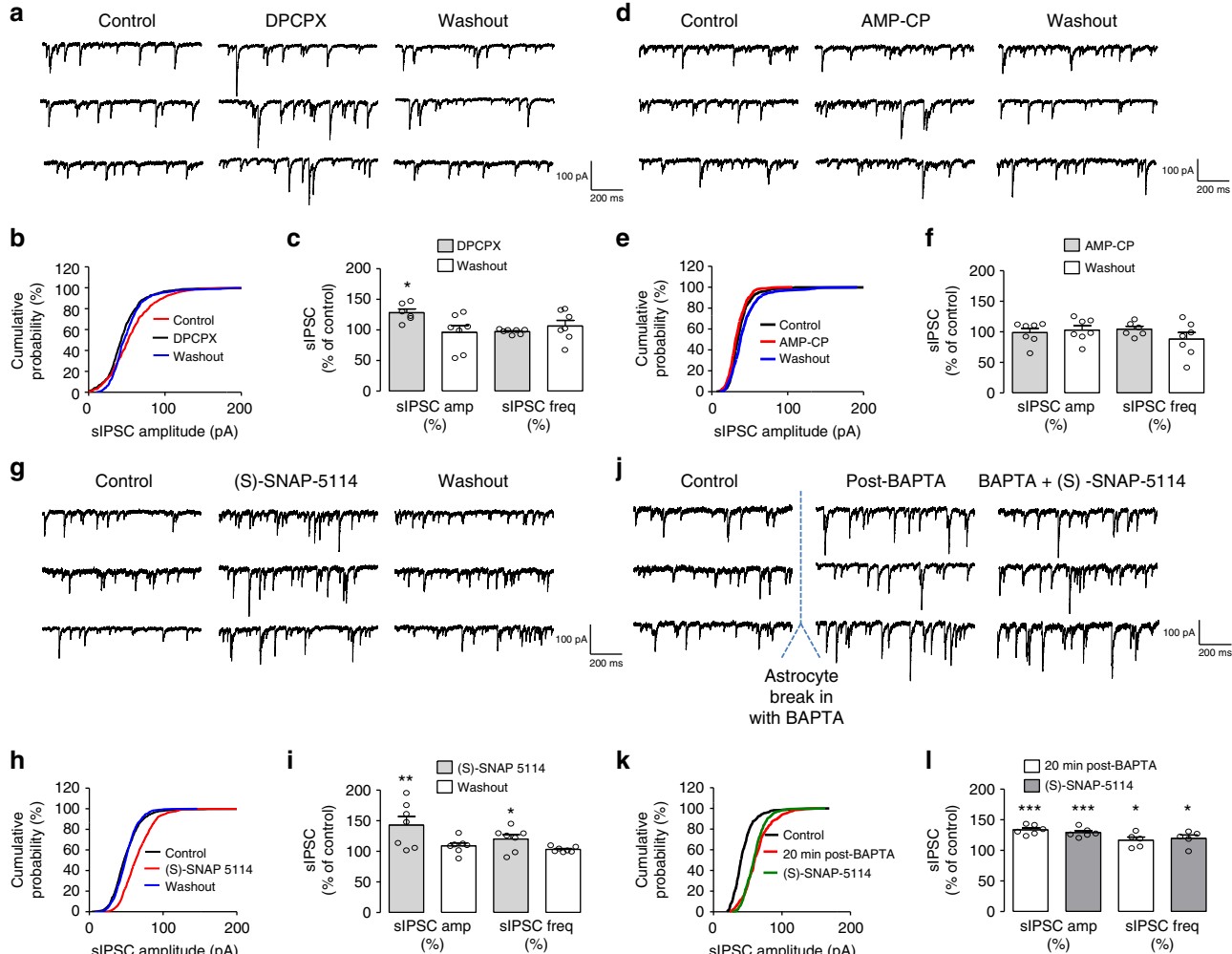

**Fig. 7** Distinct modulation of sIPSCs in pyramidal cells by $A_1$Rs, GAT-3, and astrocyte $Ca^{2+}$ activity. **a–c** Representative sIPSC traces (**a**), cumulative probability plots (**b**), and summary bar graphs (**c**) showing the increase in sIPSC amplitude, but not frequency, after 20 min application of the $A_1$R antagonist DPCPX, and the return to control after 30 min washout ($n = 7$). **d–f** Representative sIPSC traces (**d**), cumulative probability plots (**e**), and summary bar graphs (**f**) illustrating the lack of effect of 20 min application of the inhibitor of CD73/ecto-5′-nucleotidase, AMP-CP on sIPSC amplitude and frequency ($n = 7$). **g–i** Representative sIPSC traces (**g**), cumulative probability plots (**h**), and summary bar graphs (**i**) showing the increase in both sIPSC amplitude and frequency, after 20 min application of the GAT-3 inhibitor (S)-SNAP-5114, and the return to control after washout ($n = 7$). **j–l** Representative sIPSC traces (**j**), cumulative probability plots (**k**), and summary bar graphs (**l**) showing the increase in both sIPSC amplitude and frequency 20 min after whole-cell break-in and BAPTA dialysis in astrocytes, and absence of further effects on sIPSCs with additional application of (S)-SNAP-5114 ($n = 6$). *$p < 0.05$, **$p < 0.01$, ***$p < 0.001$ (see Supplementary Table 1 for detailed statistical tests)

mounted on an Zeiss LSM 510 confocal laser scanning microscope installed on a Zeiss Axioskop FS Upright Microscope (Carl Zeiss, Kirkland, Québec, Canada) and equipped with a 40× water immersion long-working distance objective (0.8 n.a.). CA1 pyramidal cells were visually identified based on their soma location and triangular shape. Astrocytes in the *stratum radiatum* (s.r.) were identified by their specific labeling with sulforhodamine 101 red fluorescent dye (SR101, 0.25 μM)[37,38]. Preliminary experiments showed that SR101 did not affect PC membrane properties and spontaneous IPSC frequency and amplitude (Supplementary Fig. 2c), unlike previously suggested for EPSCs[39]. SOM-INs and PV-INs expressing ChR2 were identified by specific EYFP fluorescence and soma location in CA1 *stratum oriens* or near *stratum pyramidale* respectively[25,28–32].

**Electrophysiology**. Whole-cell voltage-clamp recordings of CA1 pyramidal cells were obtained using borosilicate glass pipettes (3–5 MΩ) filled with intracellular solution containing the following (in mM): 130 CsCl, 10 NaCl, 10 HEPES, 1 EGTA, 0.1 CaCl₂, 10 creatine-PO₄ di(tris), 4 ATP-Mg, 0.4 GTP-Na and 5 lidocaine N-ethyl bromide (QX-314; voltage-gated $Na^+$ channel blocker) (pH 7.2 adjusted with CsOH; 285–290 mOsmol). Data was acquired using a Multiclamp 700B amplifier (Molecular Devices) and digitized using a Digidata 1320A digitizer and pClamp 10.3 (Molecular Devices). Recordings were low-pass filtered at 2 kHz and digitized at 20 kHz. Series resistance (Rs) was 10–25 MΩ and regularly monitored during experiments. Data were included only if the holding current and Rs were stable

(<20% change) throughout the experiment. $GABA_A$ receptor-mediated inhibitory postsynaptic currents (IPSCs) were recorded with pyramidal cells held at −60 mV ($Cl^-$ reversal potential = 0 mV) and confirmed with the antagonist Gabazine (5 μM, Sigma/Aldrich) (Supplementary Fig. 1c, d).

Whole-cell current-clamp recordings of EYFP-expressing SOM-INs and PV-INs were performed using borosilicate glass pipettes (3–5 MΩ) filled with a solution containing (in mM): 130 K-gluconate, 10 HEPES, 5 KCl, 5 NaCl, 4.0 ATP-Mg, 0.3 GTP-Na, 10 Na₂–creatine–PO₄ (pH 7.2–7.3 adjusted with KOH; 290–295 mOsmol). SOM-INs were characterized by a fast-spiking firing pattern with constant adaptation ratio upon the delivery of a suprathreshold depolarizing current.

Whole-cell current-clamp recordings of astrocytes were performed using borosilicate glass pipettes (5–7 MΩ) filled with a solution containing (in mM): 125 KMeSO₄, 10 HEPES, 4 MgCl₂, 4 ATP-Mg, 0.4 GTP-Na, 10 Na₂–creatine–PO₄, 0.1 Alexa Fluor 488 (pH 7.2–7.3 adjusted with KOH; 295–300 mOsmol), as previously[3,11]. For experiments with BAPTA tetrapotassium salt (0.1 or 20 mM, Sigma/Aldrich) the concentration of KMeSO₄ was adjusted to maintain the concentration of potassium ions[11]. Astrocytes were identified by their low membrane input resistance (4–15 MΩ), hyperpolarized resting membrane potential (~−70 to −90 mV), linear current-voltage profile (in voltage-clamp mode), lack of action potentials (see Supplementary Fig. 2b), and extensive syncytium revealed by the diffusion of Alexa Fluor 488[37]. Astrocyte recordings were kept only if resting membrane potential was stable and at least −70 mV.

This resting membrane potential approximately corresponds to the astrocyte reversal potential for Cl$^-$, ensuring that no significant net Cl$^-$ flow could affect the experiments.

**Optogenetic stimulation**. ChR2 was activated in SOM-INs and PV-INs by illumination using a light guide positioned above the CA1 area of the slice (473 nm blue light, custom-made light-emitting diode (LED) system)[61]. The measured LED power was 40 mW at the tip of a 1 mm (i.d.) light guide. For each PC, IPSCs were evoked first by light stimulation of different duration (0.5–5 ms; 0.1 Hz) to determine IPSC input–output function. Similar light stimulation evoked depolarizations and 1–2 action potentials in whole-cell current clamp recordings from SOM-INs (see Supplementary Fig. 1a, b) and 1 action potential on PV-INs. For pharmacological experiments, light stimulation (0.1 Hz) was adjusted to yield IPSCs of 30–40% of maximal amplitude. For each experiment, IPSCs were monitored during a control baseline period, after 20 min of drug application, and after 30 min of washout. For Ca$^{2+}$ transients evoked in astrocytes by optogenetic SOM-IN stimulation, trains of 5 ms pulses of light were given at 1 Hz for 5 s.

**Calcium imaging of astrocytes**. Whole-cell current-clamp recordings were obtained from SR101-positive astrocytes in acute hippocampal slices from SOM-ChR2/EYFP mice. Cells were loaded with the near-infrared Ca$^{2+}$ indicator CaSiR-1[40,41] via the patch pipette (100 μM CaSiR-1 potassium salt; Goryo Chemical, Inc, Sapporo, Japan). Ca$^{2+}$ imaging was performed with a LSM 510 confocal laser-scanning microscope and software (Carl Zeiss, Kirkland, Quebec, Canada) in the presence of AP-5 (20 μM), NBQX (10 μM), mGluR5 antagonist MPEP (25 μM), and when indicated Gabazine (5 μM). SR101 was excited with the 543 nm laser and detected using a 565–615 nm band-pass filter. CaSiR-1 was excited with the 633 nm laser (attenuated to 10–15% of maximum power) and detected using a 650 nm long-pass filter. To image Ca$^{2+}$ responses in astrocyte processes, images (256 × 256 pixels) were acquired at a rate of 5 frames/s. Fluorescence intensity was determined in individual astrocytes by measuring the average pixel values in 2–3 circular regions of interest (ROIs—2 μm diameter) placed over random proximal astrocytic processes (1–2 processes per astrocyte) and subtracted to a control extracellular background ROI. Changes in fluorescence (ΔF) were calculated as relative changes of fluorescence over baseline fluorescence and expressed as % ΔF/F = [($F_{post}$ − $F_{rest}$)/$F_{rest}$] × 100. Images were further analyzed off-line with LSM 510 (Carl Zeiss) software and Graph-pad Prism software (Version 5.0, GraphPad, USA).

**Immunohistochemistry**. SOM-ChR2/EYFP and PV-ChR2/EYFP mice (4–8-week old) were deeply anesthetized with sodium pentobarbital (i.p. 350 mg/kg; MTC Pharmaceuticals, Cambridge, Ontario, Canada) and transcardially perfused with 4% paraformaldehyde in ice-cold 0.1 M phosphate buffered saline (PBS). The brains were removed, post-fixed overnight, washed in PBS and cryo-preserved in 30% sucrose. Coronal sections (50 μm thick) were obtained using a freezing microtome (Leica SM200R), permeabilized with 0.4% or 0.3% Triton X-100 in PBS (15–30 min) and unspecific binding was blocked with 10% normal goat serum in 0.1% Triton X-100/PBS (1 h). Sections were then incubated with primary antibodies overnight at 4 °C. Antibodies used were: Rabbit polyclonal Anti-GFP (1/200, Thermoscientific #A-11122), Guinea Pig polyclonal Anti-GAT-3 (1/500, Synaptic Systems #274304), Mouse monoclonal Anti-GABA$_B$ R1 (1/400, Santa Cruz Biotechnology #sc-166408), Mouse monoclonal Anti-CAMKII-α (1/200, Thermoscientific #MA1-048), Rabbit polyclonal Anti-GFAP (1/300, Dako #Z0334), Mouse monoclonal anti-Parvalbumin (1/5000, Millipore #MAB1572), and Rabbit polyclonal Anti-S100β (1/300, Dako #Z0311). Sections were rinsed 3 × 10 min in PBS and then incubated with secondary antibodies for 90 min at RT. For the quadruple immuno-labeling with GAT-3, GABA$_B$R, GFAP, and S100β, each primary antibody was incubated individually and washed as described above. Secondary antibodies used were: Donkey Alexa Fluor 488-conjugated anti-rabbit IgGs (1/500, Thermofisher #A21206), Donkey Alexa Fluor 594-conjugated anti-guinea pig IgG (1/500, Jackson ImmunoResearch Laboratories #706-585-148), Goat Alexa Fluor 594-conjugated anti-rabbit IgG (1/500, Jackson ImmunoResearch Laboratories #111-585-003), Donkey Alexa Fluor 647-conjugated anti-mouse IgG (1/500, Thermofisher #A31571), and Goat Rhodamine-Red-X conjugated anti-mouse IgG1 (1/200, Jackson ImmunoResearch Laboratories #115-295-205). Sections were rinsed, mounted with Vectashield mounting medium, and examined on epi-fluorescence or Zeiss LSM 510 confocal laser scanning microscope.

**Drugs and chemicals**. Reagents were purchased from Sigma-Aldrich, unless stated otherwise. Stock solutions were made and diluted in ACSF just before bath application. Drugs used were A$_1$R selective antagonist DPCPX (100 nM), A$_1$R selective agonist N$^6$-CPA (1 μM, Tocris Bioscience), Ecto-5′-nucleotidase/CD73 inhibitor AMP-CP (200 μM, Tocris Bioscience), GAT-3 blocker (S)-SNAP 5114 (100 μM, Tocris Bioscience), GABA$_B$R selective antagonist CGP55845A (2 μM, Tocris Bioscience), GABA$_A$R selective antagonist Gabazine (5 μM), mGluR5 selective antagonist MPEP (25 μM, Tocris Bioscience), selective calcium chelating reagent BAPTA tetrapotassium salt (0.1 or 20 mM), NMDAR antagonist AP-5 (20 μM), AMPA/kainate receptor antagonist NBQX (10 μM, Tocris Bioscience).

**Statistical analyses**. Results are presented as mean ± SEM. Data with one variable (e.g., BAPTA) were analyzed with the two-tailed Student's t-test or Mann–Whitney test. Data with more than two conditions (e.g., drugs, washout) were first screened for a Gaussian distribution with Kolmogorov–Smirnov test followed by analysis either with one-way/repeated measures ANOVA or Kruskal–Wallis/Friedman test when needed and Tukey's multiple-comparison parametric post hoc test (data with Gaussian distribution) or by a Dunn's multiple-comparison non-parametric post hoc test (data with non-Gaussian distribution). Graphic significance levels were *$p < 0.05$; **$p < 0.01$ and ***$p < 0.001$. All data were analyzed using GraphPad Prism software (Version 5.0, GraphPad, USA).

## Data availability

The datasets that support the findings of this study are available from the corresponding author upon reasonable request.

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

## Acknowledgements

This work was supported by grants from the Natural Sciences and Engineering Research Council of Canada (NSERC Discovery Group grant) and an infrastructure grant from the Fonds de la Recherche du Québec—Santé (Groupe de Recherche sur le Système Nerveux Central; GRSNC) to J.-C.L and R.R. J.-C.L. is the recipient of the Canada Research Chair in Cellular and Molecular Neurophysiology. M.M. and C.R. were supported by EMBO long-term scholarships for postdoctoral research (ALTF 453-2014, ALTF 1236-2014, respectively), and I.R. by postdoctoral fellowships from GRSNC (Herbert Jasper fellowship) and Savoy Foundation. We thank Dr. H. Darabid for helpful discussions and suggestions.

## Author contributions

J-C.L., R.R., and M.M. designed the project. A.P. contributed to the experimental design and data interpretation. M.M. performed electrophysiological experiments, $Ca^{2+}$ imaging, immunohistochemistry and data analysis. A.B. performed electrophysiological experiments and data analysis. I.R. performed electrophysiological experiments on somatostatin interneurons. C.R. assisted performing control experiments with SR101 in Supplementary Figures. J.V. and I.L assisted with technical support. M.M., A.B., J-C.L. and R.R. wrote the paper. M.M. designed Figs. 1a, 2a, 3a, and 5g. A.B. designed Fig. 6a. All authors discussed the results and commented on the manuscript.

## Additional information

**Competing interests:** The authors declare no competing interests.

