## [Peer Review File · Nature Communications]

Reviewers' comments:

Reviewer #1 (Remarks to the Author):

In this manuscript, Matos and colleagues describe that the amplitude of SOM-IN to PYR inhibitory post-synaptic currents is enhanced by A1Rs in the hippocampal CA1 stratum radiatum. Using mouse hippocampal slices, they showed that GABA released by SOM-INs elicits Ca²⁺ elevations in astrocytes via the GAT-3 GABA transporter. The elevated Ca²⁺ triggers the release of ATP from astrocytic processes. The ATP in the extracellular space is converted to adenosine which in turn activates A1Rs. While the finding is potentially interesting and novel in that synaptic transmission of a specific class of interneuron is modulated by astrocytes, the dynamical it remains unclear how this circuit dynamically operates and contributes to the function of the hippocampus. For instance, is the SOM-IN to Pyr IPSC enhancement tonic or phasic? If latter, what is the time course of enhancement/decay since astrocytic Ca²⁺ elevation? How does it contribute/influenced to/by synaptic plasticity or hippocampal LFP events (ripples, sharp waves, theta, gamma, etc.)? In addition, there are a few points that need to be resolved to assure that the authors' proposal.

It remains unclear whether SOM-INs-driven Ca²⁺ elevations in astrocytes enhance SOM-INs-evoked IPSCs in pyramidal neurons, because BAPTA dialysis or GAT-3 inhibition reduces Ca²⁺ activities in wide-spread astrocytes. As it stands, endogenous activities of other GABAergic cell activity could also activate astrocytes through GAT-3 and/or GABA-BRs. In the light of the recent Tan et al. paper (doi: 10.1038/ncomms13772), ATP can excite CCK interneurons which are abundant in the CA1 stratum radiatum (e.g. Whissell et al. doi: 10.3389/fnana.2015.00124). The CCK-IN – astrocyte coupling be a possible mechanism to sustain tonic levels of GABA and ATP/adenosine. Does selective SOM-INs activation or inhibition induce enhancement or reduction of SOM-IN-evoked IPSCs, respectively?

Astrocytic Ca²⁺ elevations occur seconds after optogenetic SOM-IN activation. If endogenous SOM-IN activity sets the extracellular adenosine tone, what is the spontaneous firing rate of SOM-INs in the current experimental condition (i.e. with AP5 and NBQX)? Similarly, spontaneous astrocytic Ca²⁺ activity should be quantified.

The idea of distinct regulation of dendritic SOM-INs-evoked IPSCs relies on the comparison between SOM-INs-evoked IPSCs and spontaneous IPSCs which reflect perisomatic inhibition. Recording of IPSCs evoked by optogenetic activation of perisomatically targeting interneurons enables a fair comparison, which makes this interesting idea more acceptable.

The mechanism for the A1R-dependent IPSC enhancement is not clear. How does the author conclude the role of post-synaptic A1R while (s)-SNAP-5114 would block astrocytic, pre-, and post-synaptic A1Rs. Did the authors measure the paired pulse ratio for SOM-IN – PYR synaptic transmission?

One of the growing concerns in the community is the BAPTA-leakage problem: If BAPTA can spread to gap junction-connected neighbouring astrocytes, it could as well leak to the

extracellular space via hemichannels. The authors should make sure that BAPTA does not leak to the extracellular space. If it did, the high concentration of BAPTA (20 mM) would have a serious impact on extracellular Ca²⁺ and hence synaptic transmission. (The increased IPSC amplitude in Figs 6j-l may argue against BAPTA leak. However, most of the IPSCs are of perisomatic origin which may be out of reach of BAPTA diffusion and might reflect some homeostatic IPSC changes.)

Suppl. Fig. 3: The correctness of GAT-3, GABAB-R, and A1R IHC needs to be demonstrated (or cite the literature that uses the respective antibodies; GAT-3 and GABAB-R should not be abundant in the liver and lung.).

Minor points

Scales in micrographs $\mu\text{m}::\mu\text{M}$

Lines 268-270 & 297-299: These statements must be supported in a quantitative manner. It seems that any axons in the hippocampus is close to GAT-3 given the astrocytic coverage of the brain and the frequent punctate distribution of GAT-3.

Fig 3d: scale bar is missing. (assumed to be the same as 3j)

Fig. 3k, the authors should try 2-way ANOVA to assess the interactive effects of SNAP and CGP. [Optional: likewise, would combined application of SNAP and CGP further attenuate SOM-IN – PYR IPSC?]

Fig 6e,f: the magnified area in f looks much smaller than the dashed square region in e.

More detailed analyses are needed for Ca²⁺ imaging of astrocytes. Although "optogenetic stimulation of SOM-IN-induced Ca²⁺ transients in all astrocytic processes analysed (n=24)", the current manuscript does not clearly described how ROIs were selected. How similar/different are opto-SOM-IN-induced Ca²⁺ transients to/from spontaneously-observed ones? How does the Ca²⁺ length change with stimulus strength? How does the optogenetic excitation of SOM-INs compare with known firing rates of SOM INs in literature? A spatial analysis of astrocytic Ca²⁺ elevation in relation to SOM-INs axon terminals should also be performed.

Shigetomi et al.'s (2012; doi:10.1038/nn.3000) finding that GAT-3 surface expression can be rapidly modulated by astrocytic BAPTA dialysis could be mentioned.

Reviewer #2 (Remarks to the Author):

In this study the authors focus on somatostatin expressing interneurons (SOM-IN) and the ability of an astrocytic source of adenosine to modulate this pathway. The experiments that are performed utilize the selective expression of channelrhodopsin in SOM IN to permit selective activation of this specific sub-class of interneuron. The authors report that astrocytes detect SOM-IN activity through a combination of GABA_BRs and GAT-3 dependent Ca²⁺ signaling mechanisms and that this leads to triggering the release of ATP which is hydrolysed to adenosine which in turn causes an A1R upregulation of SOM-IN IPSCs.

This study is extremely interesting and provides novel insights into interactions between a specific sub class of interneuron and astrocytes. However, the studies are too preliminary in terms of understanding the GAT-3 mechanism as well as superficial in terms of target identification. Additional studies, including the use of A1R, CD73 and GAT-3 KO mice, are required to fully examine the proposed pathway.

P7: Are the effects of AMP-CP prevented by the A1R antagonist DPCPX as would be predicted? Are the effects of adenosine on the SOM IPSCs mediated pre or postsynaptically?

P8: Important experiments are performed using BAPTA to prevent changes in astrocytic Ca²⁺. The effects of 01 versus 20mM BAPTA are compared. Were these solutions isosmotic given the great sensitivity of astrocytes to small osmotic changes?

P9: GAT-3 immunoreactivity is presented which is appropriate. It is also important to examine existing databases of cell specific transcriptome studies to confirm that those studies demonstrate expression of the mRNA, and to cite accordingly.

On page 9 the authors introduce the idea of GABA transporters by pharmacologically interfering with GAT-3. This is important work. Presumably GAT-3 was focused on because of its enrichment in astrocytes. However, the work would be augmented by i) using antagonists to other GABA transporters and determining whether effects are specific to GAT-3, and ii) by using GAT-3 KO mice to confirm on target activity of the GAT-3 antagonist. These additional studies are appropriate both in the Ca²⁺ and synaptic modulation components of the ms.

P10 last line: A lack of statistical difference is concluded but only an N=4 is used. This is often an insufficient N number to observe statistically significant differences which emerge as larger sample sizes are examined. Please perform additional studies to determine whether there is indeed no difference. With further n values I anticipate statistical significance will be reached which would significantly change some of the conclusions.

P11: Studies are consistent with the GAT-3 being important in controlling the A1R modulation of SOM-IN inhibition of pyramidal neurons. However, all A1R evidence in the ms is weak, relying on a single dose of the antagonist DPCPX. More effort should be taken into clearly establishing the role of A1R. This evidence should include the use of A1R agonists, as well as A1R knockout mice.

P12: Immuno co-localization is presented to show a correlation between the localization of GAT-3 and A1Rs. However, for this evidence to be more compelling quantitative approaches are necessary.

P13: AMP-CP is used to examine the role of CD73 in mediating the hydrolysis of ATP to adenosine. This should be extended by using CD73 knockout mice.

The mechanism of GAT3 inhibition leading to alterations in Ca²⁺ signals is speculative at

best and further studies are required to understand mechanism.

Figure 2d – Does an A1R agonist rescue the BAPTA induced reduction of the IPSC? Similarly, one would predict rescue by ATP and that these effects would be prevented by A1R antagonists and in A1R KO mice.

Figure 4g – Does A1R agonist rescue the IPSC magnitude in the presence of SNAP and BAPTA? Is the effect of A1R agonist prevented in A1R KO mice and by DPCPX?

Reviewer #3 (Remarks to the Author):

The manuscript by Matos et al. describes a form of feedback modulation of the inhibitory events evoked by CA1 PC dendrite-innervating SOM+ IN through activation of astrocytes. The feedback is completed by the activation of A1Rs by adenosine following the conversion of ATP released by the astrocytes. The authors also claim that this effect is specific to SOM+ IN. There are two major flaws with the claims:

- 1) The comparison of the drug effects onto sIPSCs and those evoked by optogenetic stimulation of SOM+ IN is equivocal. Most of the sIPSCs recorded in the somata of CA1 PCs originate not from the activity of IN innervating distant dendrites, but from the soma-targeting IN (e.g., PV+ and CCK+ basket cells). A fair comparison would have been to express ChR2 in other dendrite-targeting IN (e.g., PV+ bistratified cells, or other IN types), and to carry out a side-by-side comparison of the pharmacological effects on the two (or more) types of optogenetically evoked IPSCs. Such experiments would substantiate the claim that the modulation is unique to the SOM+ IN, which would constitute a significant finding. Otherwise the modulation of IPSCs by gliotransmitters or glial intermediaries is by itself not novel.
- 2) The experiments appear to be seriously underpowered. The n's range from 4-8 with a very few exceptions. No attempt has been made to show that the data are normally distributed (I doubt that this can be shown on datasets with an n of 4). Moreover, the t-tests are repeatedly used on normalized data, which by definition reduces the variance. At least, repeated measures ANOVA statistics should be performed when there are repeated measurements in the data sets, such as control->drug->wash.

Response to specific comments of the reviewers.

Reviewer #1: In this manuscript, Matos and colleagues describe that the amplitude of SOM-IN to PYR inhibitory post-synaptic currents is enhanced by A1Rs in the hippocampal CA1 stratum radiatum. Using mouse hippocampal slices, they showed that GABA released by SOM-INs elicits Ca²⁺ elevations in astrocytes via the GAT-3 GABA transporter. The elevated Ca²⁺ triggers the release of ATP from astrocytic processes. The ATP in the extracellular space is converted to adenosine which in turn activates A1Rs. While the finding is potentially interesting and novel in that synaptic transmission of a specific class of interneuron is modulated by astrocytes, the dynamical it remains unclear how this circuit dynamically operates and contributes to the function of the hippocampus.

1- For instance, is the SOM-IN to Pyr IPSC enhancement tonic or phasic? If latter, what is the time course of enhancement/decay since astrocytic Ca²⁺ elevation? How does it contribute/influenced to/by synaptic plasticity or hippocampal LFP events (ripples, sharp waves, theta, gamma, etc.)?

We share the reviewer's interest for these unresolved questions. In the present study, we aimed first at establishing and deciphering how astrocytes could interact with interneurons to regulate inhibition of excitatory cells. and shape the network function in the hippocampus. Investigating how, as a consequence, these mechanisms shape the network function in the hippocampus is important, but it represents a substantial amount of work that is beyond the scope of the present paper. Nevertheless we added a comment in the discussion section, line 492-496 to address these points.

“At another level of regulation, astrocytes are able to influence rhythmic firing of neurons (Morquette & Tan), therefore it would also be relevant to investigate the relationship between astrocyte and inhibitory synapses in the modulation of rhythmic brain activities that are important for hippocampal functions.”

2- In addition, there are a few points that need to be resolved to assure that the authors' proposal.

It remains unclear whether SOM-INs-driven Ca²⁺ elevations in astrocytes enhance SOM-INs-evoked IPSCs in pyramidal neurons, because BAPTA dialysis or GAT-3 inhibition reduces Ca²⁺ activities in wide-spread astrocytes. As it stands, endogenous activities of other GABAergic cell activity could also activate astrocytes through GAT-3 and/or GABA-BRs.

To address this point, we performed additional experiments with PV-ChR2/EYFP transgenic mice and found that bath application of the GAT-3 specific antagonist (S)-SNAP-5114 (10 μM) did not affect the amplitude of PV-IPSCs in pyramidal cells (95.3 ± 9.1% of control, n = 6, p > 0.05, Friedman test; Fig. 6e, f). These results show that inhibition by at least another major GABAergic cell type (PV-IN) in the hippocampus is not modulated through GAT-3 activation of astrocytes. We have also modified discussion section accordingly.

3- In the light of the recent Tan et al. paper (doi: 10.1038/ncomms13772), ATP can excite CCK interneurons which are abundant in the CA1 stratum radiatum (e.g. Whissell et al. doi:

10.3389/fnana.2015.00124). The CCK-IN – astrocyte coupling be a possible mechanism to sustain tonic levels of GABA and ATP/adenosine. Does selective SOM-INs activation or inhibition induce enhancement or reduction of SOM-IN-evoked IPSCs, respectively?

As mentioned by reviewer #1, Tan *et al.*, have shown that optogenetic astrocyte stimulation increase CCK interneurons excitability via ATP and decrease pyramidal cells excitability via adenosine. We acknowledged and commented on these points in the discussion section line 451-454

“Nevertheless, it has been recently shown that optogenetic activation of hippocampal astrocytes increases the firing frequency of cholecystinin expressing interneurons (CCK-INs) via ATP release but not PV-INs and decrease pyramidal cells excitability via adenosine 58.”

In regard to the last statement of the reviewer, in our study we show that IPSCs evoked by selective optogenetic activation of SOM-INs via are enhanced through activation of GAT-3 on astrocytes and A₁R on pyramidal cells. These conclusions are detailed line 140 to 167 and Fig. 1; line 244 to 259 and Fig. 4; line 269 to 292 and Fig. 5 in the results section and in the discussion. We also show in Additional Data Fig 1, that SOM-IN firing and SOM-IPSCs increase as a function of optogenetic stimulus duration. With this experimental paradigm using optogenetic activation of SOM-INs to elicit IPSCs, it is not possible to use optogenetic inactivation of SOM-INs at the same time. But anyway our results clearly show that the level of selective SOM-INs activation results in proportional SOM-IN-evoked IPSCs and we do not feel that additional experiments are necessary to confirm this point.

4- Astrocytic Ca²⁺ elevations occur seconds after optogenetic SOM-IN activation. If endogenous SOM-IN activity sets the extracellular adenosine tone, what is the spontaneous firing rate of SOM-INs in the current experimental condition (i.e. with AP5 and NBQX)? Similarly, spontaneous astrocytic Ca²⁺ activity should be quantified.

In our conditions (i.e. with AP5 and NBQX), the spontaneous firing rate at resting membrane potential of SOM-IN is around 3-7 Hz as described in Chittajallu R *et al.*, 2013 and Amilhon B *et al.*, 2015. As pointed out by the reviewer, at present it is not known if SOM-IN activity sets the extracellular adenosine tone or if it is influenced solely by spontaneous astrocytic Ca²⁺ activity, or a combination of both.

The spontaneous astrocytic Ca²⁺ activities in the *stratum radiatum* of the hippocampus without AP5 and NBQX has been shown to occur approximately in 49 % of the astrocytic network with a mean frequency of 0.49 event/min and in 57.9 % of the microdomains with a mean frequency of 0.59 event/min as described by Nakayama R *et al.*, 2016, Rungta RL *et al.*, 2016 and Bosson A *et al.*, 2017. Spontaneous astrocytic Ca²⁺ activities in the stratum radiatum are mainly mediated by influx of extracellular Ca²⁺ and internal store opening rather than AMPA-R and NMDA-R. As a consequence, in presence of AP5 and NBQX, these parameters should reflect astrocytes Ca²⁺ activity.

In addition, each astrocyte could contact hundreds to thousands of synapses, both excitatory and inhibitory. This represents as much different inputs that could simultaneously modulate astrocyte Ca^{2+} activities at the network and microdomains levels. At present we feel that determining how these particular spontaneous Ca^{2+} activities contribute to the SOM-IN synapse regulation would require lots of experimentation and is beyond the scope of the present paper.

Thus given the uncertainty about the mechanisms responsible for extracellular adenosine tone, we preferred to refrain from discussing speculations.

5- The idea of distinct regulation of dendritic SOM-INs-evoked IPSCs relies on the comparison between SOM-INs-evoked IPSCs and spontaneous IPSCs which reflect perisomatic inhibition. Recording of IPSCs evoked by optogenetic activation of perisomatically targeting interneurons enables a fair comparison, which makes this interesting idea more acceptable.

We agree with the reviewer and to address this question we generated PV-ChR2 mice to selectively activate parvalbumin expressing interneurons (PV-INs), known to target the perisomatic area of pyramidal cells (*Kenneth A et al., 2017*). We tested if PV-IN inhibition of pyramidal cells was sensitive to endogenous AIR activation or GAT-3 activity and found it was not. We have modified the manuscript in the light of these new experiments:

Results section, line 302-324

“Endogenous activation of AIR and astrocytic GAT-3 do not regulate inhibition of pyramidal cells by PV-INs.

We next examined if AIR- and GAT-3-mediated astrocytic modulation of synaptic inhibition of pyramidal cells is specific to inhibition by SOM-INs or also regulate inhibition by other interneurons types. We targeted ChR2 expression to another major type of interneurons, the parvalbumin-expressing interneurons (PV-INs) and recorded IPSCs evoked in CA1 pyramidal cells of PV-ChR2/EYFP transgenic mice by optogenetic stimulation (Fig. 6a-b). Optogenetic stimulation of PV-INs with light pulses of different duration (0.4 to 1 ms; 0.1 Hz) evoked IPSCs (PV-IPSCs) of increasing amplitude in pyramidal cells (Extended Data Fig. 1d) that were GABAAR-mediated (Extended Data Fig. 1f).

Next we used a similar pharmacological approach as previously to determine if endogenous activation of AIRs regulates inhibition of pyramidal cells by PV-INs. Bath application of the AIR antagonist DPCPX (100 nM) failed to affect the amplitude of PV-IPSCs ($99.9 \pm 6.0\%$ of control, $n = 7$, $p > 0.05$, Fig. 6d, f). These results indicate that PV-IPSCs in pyramidal cells are not subject to regulation by endogenous adenosine activation of AIRs. We next assessed if astrocytic GAT-3 activation regulates inhibition of pyramidal cells by PV-INs using the GAT-3 specific antagonist (S)-SNAP-5114 as previously. Bath application (S)-SNAP-5114 (10 μM) did not change the amplitude of PV-IPSCs ($95.3 \pm 9.1\%$ of control, $n = 6$, $p > 0.05$, Fig. 6e, f). These results indicate that inhibition of pyramidal cells by PV-INs is unaffected by the blockade of either GAT-3 or AIR suggesting that AIR- and GAT-3-mediated astrocytic regulation of synaptic inhibition of pyramidal cells may be specific to inhibition by SOM-INs.”

We have also added a new figure (Fig 6) to illustrate these data.

Matos et al. Figure 6.

6- The mechanism for the A1R-dependent IPSC enhancement is not clear. How does the author conclude the role of post-synaptic A1R while (s)-SNAP-5114 would block astrocytic, pre-, and post-synaptic A1Rs. Did the authors measure the paired pulse ratio for SOM-IN – PYR synaptic transmission?

We agree with the reviewer that we did not address experimentally this point in the present manuscript. We did not assess changes in paired pulse ratio of IPSCs because we found an absence of paired pulse modulation with paired pulse optogenetic stimulation (IPSC2 = IPSC1). We acknowledged in the discussion (line 510-515 & 520-523) that we could not distinguish between pre- and post-synaptic mechanisms, but we acknowledge previous work of others (references 58 59 60 62) that have shown a postsynaptic mechanism involving

inhibition of postsynaptic adenylate cyclase, reduced PKA activity and increased postsynaptic GABA_AR function.

7- One of the growing concerns in the community is the BAPTA-leakage problem: If BAPTA can spread to gap junction-connected neighbouring astrocytes, it could as well leak to the extracellular space via hemichannels. The authors should make sure that BAPTA does not leak to the extracellular space. If it did, the high concentration of BAPTA (20 mM) would have a serious impact on extracellular Ca²⁺ and hence synaptic transmission. (The increased IPSC amplitude in Figs 6j-l may argue against BAPTA leak. However, most of the IPSCs are of perisomatic origin which may be out of reach of BAPTA diffusion and might reflect some homeostatic IPSC changes.)

As suggested by the reviewer, the BAPTA-leakage is important to take into account. As a consequence we and others previously carried out experiments with a pipette containing BAPTA in the extracellular space to rule out potential effects of BAPTA leakage (*Serrano A et al., 2006, Panatier A et al., 2011, Martin-Fernandez M, et al., 2017*). Thus as previously shown, in our experimental conditions the effects of BAPTA dialysis in astrocytes are unlikely to be due to BAPTA-leakage.

8- Suppl. Fig. 3: The correctness of GAT-3, GABAB-R, and A1R IHC needs to be demonstrated (or cite the literature that uses the respective antibodies; GAT-3 and GABAB-R should not be abundant in the liver and lung).

Due to Nature Communications guidelines and references limitations, we did not cite the original controls. Nevertheless, in the Methods section we provide catalog numbers and suppliers, to which readers may refer for controls for all the antibodies used in the manuscript.

GAT-3: line 636 “*Anti-GAT-3 (1/500, Synaptic Systems #274304)*”

GABAB-R: line 637-638 “*Anti-GABAB R1 (1/400, Santa Cruz Biotechnology #sc-166408)*”

A1R: line 638 “*Anti-A1R (1/100, Abcam #ab3460)*”

Minor points

9- Scales in micrographs μm :: μM

Corrected

10- Lines 268-270 & 297-299: These statements must be supported in a quantitative manner. It seems that any axons in the hippocampus is close to GAT-3 given the astrocytic coverage of the brain and the frequent punctate distribution of GAT-3.

These statements refer to the immunohistochemical data that we obtained and that provides qualitative evidence of co-localization at the light microscope level to support “*putative mechanism involving GAT-3-dependent activation of a Ca²⁺-dependent astrocytic purinergic regulation of SOM-IN inhibitory synapses on pyramidal cell dendrites*”. A detailed quantification of GAT-3, GFAP, S100B, SOM-IN-YFP and A₁R co-localization is not necessary to make our points. In addition such quantification would require extensive analysis

(deconvolution methods, super or pseudo super-resolution methods such as *airy-scan* or even electron microscopy studies) and are beyond the scope of the present paper.

11- Fig 3d: scale bar is missing. (assumed to be the same as 3j)

The scale bar is the same for 3d & e, but only present on 3e to avoid redundancy,

12- Fig. 3k, the authors should try 2-way ANOVA to assess the interactive effects of SNAP and CGP. [Optional: likewise, would combined application of SNAP and CGP further attenuate SOM-IN – PYR IPSC?]

As the drugs SNAP and CGP are considered to be the same factor, one-way ANOVA followed by Tukey's multiple comparison post hoc test is relevant. To this end, the interactive effect has been assessed by the combination of SNAP + CGP, which is significantly different (higher reduction) from SNAP and CGP alone.

13- Fig 6e,f: the magnified area in f looks much smaller than the dashed square region in e.

Here we assume that you mean Fig 4. There was a mistake on Fig 4. That we corrected with smaller dashed square regions.

14- More detailed analyses are needed for Ca²⁺ imaging of astrocytes. Although "optogenetic stimulation of SOM-IN-induced Ca²⁺ transients in all astrocytic processes analysed (n=24)", the current manuscript does not clearly describe how ROIs were selected. How similar/different are opto-SOM-IN-induced Ca²⁺ transients to/from spontaneously-observed ones? How does the Ca²⁺ length change with stimulus strength? How does the optogenetic excitation of SOM-INs compare with known firing rates of SOM INs in literature? A spatial analysis of astrocytic Ca²⁺ elevation in relation to SOM-INs axon terminals should also be performed.

These questions about Ca²⁺ transients were answered in point 4 above. In addition a description of how ROIs were selected was added in Methods (line 644).

"Fluorescence intensity was determined in individual astrocytes by measuring the average pixel values in defined in 2-3 circular regions of interest (ROIs – 2 μm diameter) placed over random proximal astrocytic processes (1-2 processes per astrocyte) and subtracted to a control extracellular background ROI."

Finally, related to point 10 above, a quantitative analysis of the spatial relationship between astrocyte Ca²⁺ elevation and SOM-IN axon terminals would require exhaustive anatomical experimentation which is beyond the scope of the present paper.

15- Shigetomi et al.'s (2012; doi:10.1038/nn.3000) finding that GAT-3 surface expression can be rapidly modulated by astrocytic BAPTA dialysis could be mentioned.

This was already acknowledged in the original manuscript, but to be more explicit the sentence was modified in discussion section, line 421 “This may be an auto-regulated mechanism since astrocytic Ca²⁺ signals can conversely modulate GAT-3 activity and expression²⁴”

Reviewer #2: In this study the authors focus on somatostatin expressing interneurons (SOM-IN) and the ability of an astrocytic source of adenosine to modulate this pathway. The experiments that are performed utilize the selective expression of channelrhodopsin in SOM IN to permit selective activation of this specific sub-class of interneuron. The authors report that astrocytes detect SOM-IN activity through a combination of GABA_BRs and GAT-3 dependent Ca²⁺ signaling mechanisms and that this leads to triggering the release of ATP which is hydrolysed to adenosine which in turn causes an A1R upregulation of SOM-IN IPSCs.

This study is extremely interesting and provides novel insights into interactions between a specific sub class of interneuron and astrocytes. However, the studies are too preliminary in terms of understanding the GAT-3 mechanism as well as superficial in terms of target identification. Additional studies, including the use of A1R, CD73 and GAT-3 KO mice, are required to fully examine the proposed pathway.

1- P7: Are the effects of AMP-CP prevented by the A1R antagonist DPCPX as would be predicted? Are the effects of adenosine on the SOM IPSCs mediated pre or postsynaptically?

As suggested we performed new experiment to test if DPCPX actions occlude the effect of AMP-CP. As predicted, AMP-CP failed to produce an effect in the presence of DPCPX (66.5 ± 5.9 % of control, n = 6, p < 0.01 relative to control, p > 0.05 relative to DPCPX, ANOVA; Fig. 1f, i). Figure 1 was modified to illustrate these results (panel f & i).

About pre- or postsynaptic adenosine mechanisms, this is the same point as reviewer 1 point 6. We agree with the reviewer that we did not address this point experimentally in the present manuscript. We acknowledged in the discussion (line 510-511 & 521-524) that we could not distinguish between pre- and post-synaptic mechanisms, but previous work of others (ref 60) have shown a postsynaptic mechanism involving inhibition of postsynaptic adenylate cyclase, reduced PKA activity and increased postsynaptic GABA_AR function

Matos et al. Figure 1.

2- P8: Important experiments are performed using BAPTA to prevent changes in astrocytic Ca²⁺. The effects of 01 versus 20mM BAPTA are compared. Were these solutions isosmotic given the great sensitivity of astrocytes to small osmotic changes?

As stated in the Methods section, intracellular solutions are adequately calculated/adjusted to be isosmotic and the osmolarity was checked every time. “Whole cell current-clamp recordings of astrocytes were performed as previously using borosilicate glass pipettes (5-7 MΩ) filled with a solution containing (in mM): 125 KMeSO₄, 10 HEPES, 4 MgCl₂, 4 ATP-Mg, 0.4 GTP-Na, 10 Na₂-creatine-PO₄, 0.1 Alexa Fluor 488 (pH 7.2-7.3 adjusted with KOH; 295-300 mOsmol), as previously^{3,11}. For experiments with BAPTA tetrapotassium salt (0.1 or 20 mM, Sigma/Aldrich) the concentration of KMeSO₄ was adjusted to maintain the concentration of potassium ions¹¹.”

In addition, we know from our previous study (reference 11) that these different concentrations do not alter the astrocytic properties. “*Finally, all criteria used to monitor astrocyte properties revealed that the cells were healthy and their basic properties were unaltered by BAPTA. These include membrane resistance, the stability of the whole-cell recordings and fine morphological structures such as the shape and size of the compartments that were monitored throughout the entire length of the experiments.*” from *Panatier et al., Cell 2011* (ref #11).

3- P9: GAT-3 immunoreactivity is presented which is appropriate. It is also important to examine existing databases of cell specific transcriptome studies to confirm that those studies demonstrate expression of the mRNA, and to cite accordingly.

In agreement with the reviewer comment, we added the citation of the recently published paper by Ghirardini E, *et al.*, 2018. In this paper, the authors showed functional expression of GAT-3 in hippocampal astrocytes by using whole-cell patch-clamp and single-cell reverse transcription-PCR. The reference has been added in Results section line 217, Discussion section line 406.

4- On page 9 the authors introduce the idea of GABA transporters by pharmacologically interfering with GAT-3. This is important work. Presumably GAT-3 was focused on because of its enrichment in astrocytes. However, the work would be augmented by i) using antagonists to other GABA transporters and determining whether effects are specific to GAT-3,

We chose to focus on GAT-3 because it's the only GABA transporter known to be specific to astrocytes (whereas GAT-1 or GlyTs are also expressed in neurons). Due to this specificity of expression we also take advantage of using (S)-SNAP-5114 which is a competitive GAT-3 specific antagonist, allowing us to target specifically and reversibly astrocytes, That in itself is quite unique in the glial field in terms of specificity. Even if GAT-1 and GAT-3 could be involved in the regulation of extracellular level of GABA, only GAT-3 has been implicated by other groups in Ca²⁺ signaling in astrocytes as stated in Discussion section, line 409 to 414.

Thus we feel that using antagonists of other transporters would not alter our conclusion that the astrocyte specific GABA transporter (GAT-3) is involved in the detection and upregulation of inhibitory transmission by somatostatin interneurons onto pyramidal cells and that these additional experiments are not necessary.

5- and ii) by using GAT-3 KO mice to confirm on target activity of the GAT-3 antagonist. These additional studies are appropriate both in the Ca²⁺ and synaptic modulation components of the ms.

We are looking in this study at a hippocampal specific acute effect in mature brain, and so take advantage of the pharmacological selectivity of (S)-SNAP-5114 to GAT-3. A Cre-lox strategy for GAT-3 KO is impossible since we already use SOM Cre mice to target Chr2. Also, the use of a general KO targeting GAT-3 would impact the whole brain, both during

and after development. Moreover the generation and characterization of these mice would take a substantial amount of time beyond the scope of the present paper.

However : 1) we carried out additional experiments with another mouse line to express ChR2 in a parvalbumin expressing interneurons to show that the GAT-3-mediated modulation is not present at parvalbumin interneuron synapses; 2) we performed new pharmacological experiments showing that application of an A1R agonist reverses the modulation of IPSCs produced by GAT3 inhibition; and 3) we carried out other additional experiments to increase N numbers in instances requested by the reviewers (eg. point 6 below).

6- P10 last line: A lack of statistical difference is concluded but only an N=4 is used. This is often an insufficient N number to observe statistically significant differences which emerge as larger sample sizes are examined. Please perform additional studies to determine whether there is indeed no difference. With further n values I anticipate statistical significance will be reached which would significantly change some of the conclusions.

As requested, we performed additional experiments and our conclusions remained unchanged. With $n = 7$ for the vehicle and CGP55845 experiments, the results are as previously, inhibition of GAT-3 but not GABA_BR regulates SOM-INs inhibition of pyramidal cells via astrocyte Ca²⁺ signaling. The legend of Fig. 4 and the corresponding text in the Results section have been modified.

Line 252 to 256: *“In contrast, vehicle treatment (Fig. 4a) or bath application of the GABABR antagonist CGP55845 (Fig. 4c) did not affect the amplitude of SOM-IPSCs in pyramidal cells ($107 \pm 5.0\%$ and $92.2 \pm 10.8\%$ of control respectively, $n = 7$ each; Fig. 4d).”*

7- To avoid redundancy in our answers, we grouped the following points made by reviewer #2

P11: Studies are consistent with the GAT-3 being important in controlling the A1R modulation of SOM-IN inhibition of pyramidal neurons. However, all A1R evidence in the ms is weak, relying on a single dose of the antagonist DPCPX. More effort should be taken into clearly establishing the role of A1R. This evidence should include the use of A1R agonists, as well as A1R knockout mice.

Figure 2d – Does an A1R agonist rescue the BAPTA induced reduction of the IPSC? Similarly, one would predict rescue by ATP and that these effects would be prevented by A1R antagonists and in A1R KO mice.

Figure 4g – Does A1R agonist rescue the IPSC magnitude in the presence of SNAP and BAPTA? Is the effect of A1R agonist prevented in A1R KO mice and by DPCPX?

We share the reviewer concerns about the need for direct evidence of A₁R agonist. Regarding the use of A₁R KO mice, we feel the effects will be too extensive and deleterious since those KO are known to deeply impact synaptic transmission and induce defects in excitatory

transmission, lower seizure thresholds and cognitive/anxiogenic dysfunctions (Johansson et al., 2001; Fredholm et al., 2005 ; Fedele *et al.*, 2006 Kochanek *et al.*, 2006). Strength of the present study is the ability to transiently modulate the network with the use of specific pharmacology in paired experiments where each cell is its own control.

But as requested by the reviewer we performed additional rescue experiments with a specific A₁R agonist. We now report that N⁶-CPA rescues SNAP-5114 reduction of IPSCs and this effect is blocked by the A₁R antagonist.

As a consequence the following paragraph and figure has been added in the Results sections line 280 to 292: (note that the effect of N⁶-CPA alone is not shown in the manuscript as described below but shown to the reviewer after figure 5)

“To further confirm this mechanism we tested if bath application of the A₁R agonist N6-cyclopentyladenosine (N6-CPA, 1 μM) during the blockade of GAT-3 could up-regulate inhibition of pyramidal cells by SOM-INs. Under normal basal conditions (in absence of inhibitors), bath application of N6-CPA did not affect the amplitude of SOM-IPSCs evoked by optogenetic stimulation (99.0 ± 10.1% of control, n = 5, p = 0.33 Wilcoxon signed-rank test; data not shown). However in the presence of the GAT-3 inhibitor (S)-SNAP-5114 that decreased amplitude of SOM-IPSCs (59.3 ± 5.9 % of control, n = 4, p < 0.001 relative to control), application of the A₁R agonist N6-CPA increased the amplitude of SOM-IPSCs (72.8 ± 5.9% of control, p < 0.001 relative to (S)-SNAP-5114 Fig 5b, d). Furthermore, this effect of N6-CPA was blocked by bath application of the A₁R antagonist DPCPX (Fig 5b, d). Overall, these results are consistent with a GAT-3 activation of astrocytes leading to ATP release, activation of A₁Rs and upregulation of pyramidal cell inhibition by SOM-INs.”

The following figure 5 has been modified to represent these new data (**Fig.5 b, d**)

Matos et al. Figure 5.

Bath application of N⁶-CPA at 1 μM alone had no effect on SOM-IPSCs amplitude as illustrated on the figure below (due to space limitation we chose not to include this figure on the manuscript).

This absence of further inhibition is probably due to endogenous tone of adenosine in the extracellular space.

8- P12: Immuno co-localization is presented to show a correlation between the localization of GAT-3 and A1Rs. However, for this evidence to be more compelling quantitative approaches are necessary.

(Similar to reviewer 1 point 10) In the manuscript we are careful to point out in a qualitative manner the evident co-localization revealed by immunohistochemistry to support the “*putative mechanism involving GAT-3-dependent activation of a Ca²⁺-dependent astrocytic purinergic regulation of inhibitory synapses on pyramidal cell dendrites*”. We feel that a quantification of GAT-3 and A₁R co-localization is not necessary to make our point. In addition such quantification would require extensive analysis (deconvolution methods, super or pseudo super-resolution methods such as *airy-scan* or even electron microscopy studies) and we feel these exhaustive experiments are beyond the scope of the present paper.

9- P13: AMP-CP is used to examine the role of CD73 in mediating the hydrolysis of ATP to adenosine. This should be extended by using CD73 knockout mice.

In the light of the additional experiments, suggested by the reviewers, that we carried for the revisions of the manuscript (see point 5 above) and for similar reasons pointed out above (about the effect of global deletions; time consuming nature of KO mice generation and characterization, in addition to the two different Cre mice lines used (SOM-ChR2/EYFP and PV-ChR2/EYFP) in the present study, we think that carrying out these additional experiments is beyond the scope of the present paper.

10- The mechanism of GAT3 inhibition leading to alterations in Ca²⁺ signals is speculative at best and further studies are required to understand mechanism.

We addressed this point of the reviewer by acknowledging previous published work about GAT-3 and astrocyte Ca²⁺ mechanisms in a paragraph in the discussion (line 417-425).

“As previously suggested^{17,18}, GAT-3 mediated Ca²⁺ events could involve GAT-3 activation leading to inhibition of Na⁺/Ca²⁺ exchanger and subsequent Ca²⁺-induced Ca²⁺ release from internal stores. This may be an auto-regulated mechanism since astrocytic Ca²⁺ signals can conversely modulate GAT-3 activity and expression²⁴. The possibility of a coordinated transporter/receptor mechanism involving GAT-3 and GABABRs in Ca²⁺ transients in astrocytes is also possible, owing to their intimate co-localization in astrocytes (Fig. 3 and Extended Data Fig. 4) and a recent report showing that GABABRs controls GAT-3 levels in astrocytes in vivo during synaptogenesis⁵⁷.”

Reviewer #3: The manuscript by Matos et al. describes a form of feedback modulation of the inhibitory events evoked by CA1 PC dendrite-innervating SOM+ IN through activation of astrocytes. The feedback is completed by the activation of A1Rs by adenosine following the conversion of ATP released by the astrocytes. The authors also claim that this effect is specific to SOM+ IN. There are two major flaws with the claims:

1) The comparison of the drug effects onto sIPSCs and those evoked by optogenetic stimulation of SOM+ IN is equivocal. Most of the sIPSCs recorded in the somata of CA1 PCs

originate not from the activity of IN innervating distant dendrites, but from the soma-targeting IN (e.g., PV+ and CCK+ basket cells). A fair comparison would have been to express ChR2 in other dendrite-targeting IN (e.g., PV+ bistratified cells, or other IN types), and to carry out a side-by-side comparison of the pharmacological effects on the two (or more) types of optogenetically evoked IPSCs. Such experiments would substantiate the claim that the modulation is unique to the SOM+ IN, which would constitute a significant finding. Otherwise the modulation of IPSCs by gliotransmitters or glial intermediaries is by itself not novel.

We carried out additional experiments to address the reviewer's point about the specific modulation of SOM-IN synapses by astrocytes (similar to reviewer 1 point 5). We generated PV-ChR2 mice to selectively activate parvalbumin expressing interneurons (PV-INs), known to target the perisomatic area of pyramidal cells (*Kenneth A et al., 2017*). We tested if PV-IN inhibition of pyramidal cells was sensitive to endogenous A1R activation or GAT-3 activity and found it was not. We have modified the manuscript in the light of these new experiments. These new results complement our other evidence and suggest that astrocyte-mediated GAT-3 and endogenous of A₁R activation modulate inhibitory synapses with synapse specificity, causing upregulation of SOM-IN inhibitory synapses, down-regulation of synapses for other unidentified interneurons (likely targeting perisomatic domain) and no regulation of PV-IN synapses.

We have modified the manuscript in the light of these new experiments:

Results section, line 302 to 324

“Endogenous activation of A1R and astrocytic GAT-3 do not regulate inhibition of pyramidal cells by PV-INs.

We next examined if A1R- and GAT-3-mediated astrocytic modulation of synaptic inhibition of pyramidal cells is specific to inhibition by SOM-INs or also regulate inhibition by other interneurons types. We targeted ChR2 expression to another major type of interneurons, the parvalbumin-expressing interneurons (PV-INs) and recorded IPSCs evoked in CA1 pyramidal cells of PV-ChR2/EYFP transgenic mice by optogenetic stimulation (Fig. 6a-b). Optogenetic stimulation of PV-INs with light pulses of different duration (0.4 to 1 ms; 0.1 Hz) evoked IPSCs (PV-IPSCs) of increasing amplitude in pyramidal cells (Extended Data Fig. 1d) that were GABAAR-mediated (Extended Data Fig. 1f).

Next we used a similar pharmacological approach as previously to determine if endogenous activation of A1Rs regulates inhibition of pyramidal cells by PV-INs. Bath application of the A1R antagonist DPCPX (100 nM) failed to affect the amplitude of PV-IPSCs ($99.9 \pm 6.0\%$ of control, $n = 7$, $p > 0.05$, Fig. 6d, f). These results indicate that PV-IPSCs in pyramidal cells are not subject to regulation by endogenous adenosine activation of A1Rs. We next assessed if astrocytic GAT-3 activation regulates inhibition of pyramidal cells by PV-INs using the GAT-3 specific antagonist (S)-SNAP-5114 as previously. Bath application (S)-SNAP-5114 (10 μ M) did not change the amplitude of PV-IPSCs ($95.3 \pm 9.1\%$ of control, $n = 6$, $p > 0.05$, Fig. 6e, f). These results indicate that inhibition of pyramidal cells by PV-INs is unaffected by the blockade of either GAT-3 or A1R suggesting that A1R- and GAT-3-mediated astrocytic regulation of synaptic inhibition of pyramidal cells may be specific to inhibition by SOM-INs”

We have also added a new figure to represent these data.

The discussion sections were modified accordingly.

The experiments appear to be seriously underpowered. The n's range from 4-8 with a very few exceptions. No attempt has been made to show that the data are normally distributed (I doubt that this can be shown on datasets with an n of 4). Moreover, the t-tests are repeatedly used on normalized data, which by definition reduces the variance. At least, repeated measures ANOVA statistics should be performed when there are repeated measurements in the data sets, such as control->drug->wash.

We carried out additional experiments to increase n's (Fig 4. About vehicle and CGP, we now have n = 7).

Regarding the normality of data, for each set of experiments the distribution was tested and the appropriate tests performed. Mistakes in the text about the pairing of the data were corrected. A strength of our study is that almost all the experiments concern paired data allowing us to reach significance level with the size of sampled used. The Methods and Results sections were modified as indicated below, taking into account the parametric or non-parametric type of tests and the paired or unpaired experiments.

*“Results are presented as mean \pm SEM. Data with one variable (e.g., BAPTA) were analyzed with the two-tailed Student's t test or Mann-Whitney test. Data with more than two conditions (e.g., drugs, washout) were first screened for a Gaussian distribution with Kolmogorov-Smirnov test followed by analysis either with one-way/repeated measures ANOVA or Kruskal-Wallis/Friedman test when needed and Tukey's multiple-comparison parametric post hoc test (data with Gaussian distribution) or by a Dunn's multiple-comparison non-parametric post hoc test (data with non-Gaussian distribution). Graphic significance levels were *, $p < 0.05$; **, $p < 0.01$ and ***, $p < 0.001$. All data were analyzed using GraphPad Prism software (Version 5.0, GraphPad, USA).”*

The corresponding legends for the figure were also modified accordingly.

REVIEWERS' COMMENTS:

Reviewer #1 (Remarks to the Author):

The manuscript has improved tremendously after this revision. The authors have addressed my concerns. Perhaps there is no good control experiment for BAPTA leakage. Previous control experiments of cell-attached vs. whole-cell patch clamp of astrocytes are no good, because the latter would have an easier reach of BAPTA to perisynaptic areas. The authors reported that sIPSCs were enhanced in both amplitude and frequency while evoked IPSCs from PV-INs did not change in magnitude. As such, there is a small possibility that presumed BAPTA leakage causes complex alterations to synaptic transmission. The mechanism of the enhanced sIPSCs by astrocytic BAPTA remains unknown (Fig 7), and the phenomenon appears contradictory to Fig 2 (SOM) and Fig 6 (PV, though BAPTA nor astrocytic Ca²⁺ elevation has been tested). Indeed, the mechanism could partially owe to some other types of interneurons (CCK or VIP, perhaps). Though identification of such mechanisms is beyond the scope of this manuscript, this potential predicament deserves a more detailed discussion.

Reviewer #3 (Remarks to the Author):

The authors have adequately addressed the points raised in my previous review.

Response to specific comments of the reviewers.

Reviewer #1:

The manuscript has improved tremendously after this revision. The authors have addressed my concerns. Perhaps there is no good control experiment for BAPTA leakage. Previous control experiments of cell-attached vs. whole-cell patch clamp of astrocytes are no good, because the latter would have an easier reach of BAPTA to perisynaptic areas. The authors reported that sIPSCs were enhanced in both amplitude and frequency while evoked IPSCs from PV-INs did not change in magnitude. As such, there is a small possibility that presumed BAPTA leakage causes complex alterations to synaptic transmission. The mechanism of the enhanced sIPSCs by astrocytic BAPTA remains unknown (Fig 7), and the phenomenon appears contradictory to Fig 2 (SOM) and Fig 6 (PV, though BAPTA nor astrocytic Ca²⁺ elevation has been tested). Indeed, the mechanism could partially owe to some other types of interneurons (CCK or VIP, perhaps). Though identification of such mechanisms is beyond the scope of this manuscript, this potential predicament deserves a more detailed discussion.

As suggested, we added a new paragraph in the discussion to address the problem of BAPTA leakage:

“A potential problem with the BAPTA experiments is if BAPTA spread to gap junction-connected neighbouring astrocytes it could as well leak to the extracellular space via hemichannels. Thus, in experiments with the high concentration of BAPTA (20 mM), leakage could impact on extracellular Ca²⁺ and hence synaptic transmission. However the increased IPSC amplitude in Figs 7j-l argues against such a BAPTA leakage effect. Moreover, we have previously carried out experiments with a pipette containing BAPTA in the extracellular space to rule out potential effects of BAPTA leakage^{3,11}. Thus in our experimental conditions, the effects of BAPTA dialysis in astrocytes are unlikely to be due to BAPTA leakage.”